# NÜWA: MENDING THE SPATIAL INTEGRITY TORN BY VLM TOKEN PRUNING

**Yihong Huang**[1,2], **Fei Ma**[1]*, **Yihua Shao**[3], **Jingcai Guo**[3], **Zitong Yu**[4], **Laizhong Cui**[5], **Qi Tian**[1,6]

[1]Guangdong Laboratory of Artificial Intelligence and Digital Economy (SZ)

[2]School of Artificial Intelligence, Xidian University

[3]The Hong Kong Polytechnic University [4]Great Bay University

[5]Shenzhen University [6]Huawei

`huangyihong@stu.xidian.edu.cn,mafei@gml.ac.cn`

## ABSTRACT

Vision token pruning has proven to be an effective acceleration technique for the efficient Vision Language Model (VLM). However, existing pruning methods demonstrate excellent performance preservation in visual question answering (VQA) and suffer substantial degradation on visual grounding (VG) tasks. Our analysis of the VLM's processing pipeline reveals that strategies utilizing global semantic similarity and attention scores lose the global spatial reference frame, which is derived from the interactions of tokens' positional information. Motivated by these findings, we propose Nüwa, a two-stage token pruning framework that enables efficient feature aggregation while maintaining spatial integrity. In the first stage, after the vision encoder, we apply three operations, namely separation, alignment, and aggregation, which are inspired by swarm intelligence algorithms to retain information-rich global spatial anchors. In the second stage, within the LLM, we perform text-guided pruning to retain task-relevant visual tokens. Extensive experiments demonstrate that Nüwa achieves SOTA performance on multiple VQA benchmarks (from 94% to 95%) and yields substantial improvements on visual grounding tasks (from 7% to 47%).

**Code:** `https://github.com/Man-PaperRejected/Nuwa`

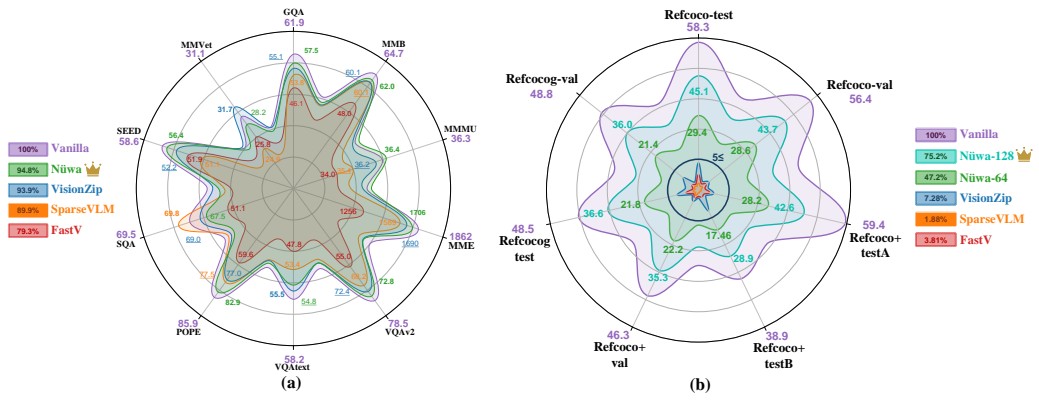

Figure 1: Nüwa Performance On VQA and VG tasks, preserving 95% and 47% under 88.9% reduction of vision tokens. (a) Our Nüwa outperforms current efficient VLMs on 10 VQA benchmarks; (b) On 3 visual grounding benchmarks, Nüwa also achieves SOTA results.

---

*Corresponding Author

# 1 INTRODUCTION

VLMs (Liu et al., 2024a; Bai et al., 2025; Zhu et al., 2025) exhibit strong multimodal capabilities through pre-training on massive image-text pairs. However, the large number of vision tokens generated during inference leads to substantial computational overhead and reduced throughput. Recent visual token pruning methods aim to accelerate inference while preserving model performance. These include approaches based on visual-semantic similarity (Yang et al., 2024; Li & Shin, 2024; Jeddi et al., 2025), textual semantic filtering (Chen et al., 2024; Xing et al., 2024; Endo et al., 2024), and multi-stage pruning (Zhang et al., 2025a; Liu et al., 2025b), which improve inference throughput. Additionally, token merging (Bolya et al., 2023), a technique within token pruning, increases token sparsity in VLMs during inference. Fundamentally, reducing the number of tokens enhances sparsity and thereby improves VLM's inference throughput.

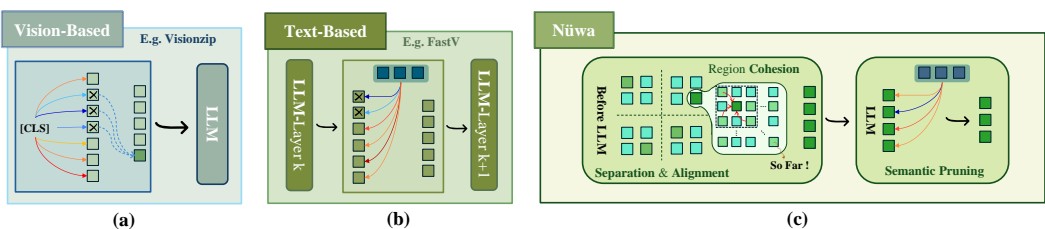

Figure 2: The left panel contrasts our Nüwa framework with prior token pruning methods. (a) Pruning at the vision encoder stage; (b) Text-guided pruning within the LLM; (c) Our two-stage approach: initial spatial-aware pruning via local aggregation that preserves global anchors in the vision encoder, followed by text-guided refinement in the LLM.

Nevertheless, recent studies (Wen et al., 2025; Endo et al., 2024) have questioned the effectiveness of existing pruning methods (Chen et al., 2024; Zhang et al., 2025b). In particular, random pruning and pooling-based merging can achieve competitive performance, yet these methods exhibit substantial degradation on visual grounding (VG) tasks compared with visual question and answering (VQA) tasks (Long et al., 2025; Shao et al., 2025a). To assess whether these issues are widespread, we systematically categorize existing pruning methods and compare them with simpler baselines across multiple datasets. Our experiments confirm that these limitations persist. These findings raise fundamental questions: ❶ *Why do existing pruning methods exhibit significant task-dependent degradation?* ❷ *How is vision information encoded and utilized within the VLM's processing pipeline?* ❸ *How to Mend grounding performance gaps in VLM's token pruning setting?*

Through systematic experimental analysis, we uncover that VLMs employ a multi-stage visual processing pipeline that progresses from global to fine-grained integration, with task-specific requirements. In particular, grounding tasks depend on preserving global spatial reference frames, which are constructed from token position information and can be disrupted by token pruning. Informed by these insights, we introduce **Nüwa**, as shown in Figure 2, a two-stage spatial-aware token pruning framework, patching up the torn spatial integrity. The first stage operates in the visual semantic space to reduce token redundancy while maintaining spatial topology. It employs a Boids-inspired algorithm (Reynolds, 1998) with three operations: (1) **Separation**: partitioning the token map into localized regions; (2) **Alignment**: selecting representative tokens based on their alignment with the global context and information density; and (3) **Aggregation**: merging features of neighboring tokens around representatives using semantic similarity. The second stage performs text-guided refinement in the intermediate layers of the LLM after multimodal feature alignment, using textual semantics to guide further pruning.

**Nüwa** demonstrates significant improvements, as shown in Figure 1, on VG benchmarks (**7%→47%, 18%→75%**) across multiple pruning configurations in LLaVA-1.5, alongside enhancements in VQA benchmarks, including image reasoning and understanding performance (**94%→95%**), and validates its effectiveness across additional models.

Our contributions are as follows:

1. **Task-specific Analysis:** We systematically examine VLM's processing pipelines and show that current pruning methods fail on grounding tasks by overlooking task-specific requirements and disrupting spatial structure. Position reconstruction experiments confirm that spatial perception arises from the integrity of the global reference frame.

2. **Nüwa Framework:** We propose a two-stage spatial-aware pruning framework that retains global spatial anchors through separation and adaptive region aggregation, thereby preserving both spatial and semantic integrity. It further leverages textual information in the LLM for multimodal alignment-based pruning.

3. **Performance Validation:** Our approach yields superior results across 13 datasets and multiple VLMs, establishing new SOTA on VQA (95% performance retention) and VG (47.2% performance retention) tasks while achieving 89% reduction in TFLOPs and 62% reduction in prefill time with a 88.9% tokens reduction.

## 2 DISSECTING THE VISUAL PROCESSING PIPELINE: FROM SEMANTIC FLOW TO SPATIAL INTEGRITY

In this section, we first perform a systematic analysis (Sec 2.1) of existing pruning methods to address two key questions. We then examine the visual information processing pipeline (Sec 2.2) in VLMs through two analytical experiments, tracing the progression from global attention mechanisms to local processing paradigms. Finally, position reconstruction experiments (Sec 2.3) uncover the root causes of performance degradation in grounding tasks, thereby providing insights for the design of pruning methods.

### 2.1 EVALUATING COMPETITIVE ADVANTAGES: SIMPLE BASELINES VERSUS ADVANCED PRUNING METHODS

Recent research (Wen et al., 2025; Endo et al., 2024) has questioned the effectiveness of existing visual token pruning methods. To investigate two key aspects — **(1) Generalization:** whether advanced methods consistently outperform simple baselines, and **(2) Robustness:** whether performance remains stable across tasks with diverse requirements, we conduct a comprehensive cross-task evaluation.

**Experimental Setup** We conduct a comprehensive evaluation across 12 datasets, covering a broad spectrum of capabilities including image grounding, fine-grained understanding, and complex reasoning. To facilitate a systematic comparison, we categorize mainstream visual token pruning methods into three distinct families based on their architectural placement and operation stage: **Vision Encoder-Side Pruning**, which focuses on reducing redundancy within or at the output of the vision encoder to save memory early on (e.g., VisionZip (Yang et al., 2024), PruMerge (Shang et al., 2024)); **LLM Single-Layer Pruning**, which applies a one-time, fixed-ratio pruning operation at specific layers within the LLM (e.g., FastV (Chen et al., 2024)); and **LLM Multi-Layer Pruning**, which dynamically identifies and removes non-essential vision tokens across consecutive LLM layers (e.g., PyramidDrop (Xing et al., 2024), SparseVLM (Zhang et al., 2025b)). To ensure a fair and rigorous assessment, we benchmark these sophisticated methods against two simple yet effective baselines, random sampling and average pooling, to determine the value of complex pruning designs.

Table 1: Performance comparison of various vision token pruning methods on LLAVA1.5-7B. Including LLM Single-Layer Pruning, LLM Multi-Layer Pruning, and Vision Encoder-Side Pruning.

| Method | Source | GQA | MMB | MMMU | MME | VQAv2 | VQAtext | POPE | SQA | MMVet | Avg (%) |
|---|---|---|---|---|---|---|---|---|---|---|---|
| Vanilla | CVPR'24 | 61.9 | 64.7 | 36.3 | 1862 | 78.5 | 58.2 | 85.9 | 69.5 | 31.1 | 100.0 |
| **FastV** | ECCV'24 | 46.1 (74.5%) | 48.0 (74.2%) | 34.0 (93.7%) | 1255 (67.4%) | 55.0 (70.1%) | **47.8** (82.1%) | 59.6 (69.4%) | 68.7 (98.8%) | **23.3** (74.9%) | **78.3** |
| Random | – | 51.2 (82.6%) | 41.8 (64.5%) | 34.1 (94.0%) | 1351 (72.6%) | 65.4 (83.3%) | 44.9 (77.1%) | 61.1 (71.1%) | 66.8 (96.1%) | 16.9 (54.3%) | 77.3 |
| Pooling | – | **52.2** (84.4%) | **48.7** (75.3%) | 34.0 (93.7%) | **1380** (74.1%) | **69.1** (88.0%) | 45.3 (77.9%) | **67.8** (78.9%) | 67.9 (97.7%) | 16.3 (52.4%) | 80.3 |
| **PDrop** | CVPR'25 | 41.9 (67.7%) | 33.3 (51.5%) | 26.5 (73.0%) | 1092 (58.6%) | 57.3 (73.0%) | 45.9 (78.9%) | 55.9 (65.1%) | 69.2 (99.6%) | 24.9 (80.1%) | 72.0 |
| **SparseVLM** | ICML'25 | **53.8** (86.9%) | **60.1** (92.9%) | **35.4** (97.6%) | **1589** (85.3%) | **68.2** (86.9%) | **53.4** (91.8%) | **77.5** (90.2%) | **69.8** (100.4%) | 24.9 (80.1%) | **90.2** |
| Random | – | 51.5 (83.2%) | 46.0 (71.2%) | 34.1 (94.0%) | 1342 (72.1%) | 67.1 (85.5%) | 46.7 (80.2%) | 71.8 (83.6%) | 68.1 (98.0%) | 23.1 (74.3%) | 82.5 |
| **VisionZip** | CVPR'25 | 55.1 (89.0%) | **60.1** (92.9%) | **36.2** (99.7%) | **1690** (90.8%) | **72.4** (92.2%) | **55.5** (95.4%) | **77.0** (89.6%) | 69.0 (99.3%) | **31.7** (101.9%) | **94.5** |
| **PruMerge+** | ICCV'25 | **55.4** (89.5%) | 59.6 (92.1%) | 35.8 (98.6%) | 1616 (86.8%) | 71.3 (90.8%) | 52.0 (89.3%) | 75.7 (88.1%) | **69.5** (100.0%) | 28.0 (90.0%) | 91.7 |
| Random | – | 54.3 (87.7%) | 51.1 (79.0%) | 34.0 (93.7%) | 1410 (75.7%) | 66.2 (84.3%) | 46.5 (79.9%) | 68.2 (79.3%) | 65.5 (94.2%) | 21.1 (68.0%) | 82.4 |
| Pooling | – | 51.5 (83.1%) | 44.4 (68.6%) | 32.1 (88.4%) | 1151 (61.8%) | 68.1 (86.8%) | 42.9 (73.8%) | 68.0 (79.2%) | 64.7 (93.1%) | 18.7 (60.1%) | 77.2 |

Our results reveal key patterns across task types. On general-purpose VQA benchmarks (Table 1), simple baselines achieve competitive performance, often matching advanced pruning methods. In contrast, on object-centric grounding tasks (Table 2), all methods show systematic

Table 2: Performance comparison on RefCOCO series datasets.

| Avg Tokens | Method | Refcoco-test | Refcoco+-testA | Refcoco+-testB | Refcocog-test |
|---|---|---|---|---|---|
| 576 | LLaVA | 58.30 | 59.43 | 38.88 | 48.50 |
| 128 | FastV | 10.34 | 8.53 | 9.83 | 8.87 |
| | SparseVLM | 6.27 | 5.79 | 4.22 | 6.35 |
| | VisionZip | 4.49 | 4.06 | 4.86 | 3.50 |
| | Pooling | 23.01 | 24.37 | 15.04 | 19.69 |
| 64 | FastV | 2.73 | 1.17 | 1.02 | 2.19 |
| | SparseVLM | 1.04 | 0.96 | 1.28 | 0.61 |
| | VisionZip | 4.04 | 3.73 | 3.86 | 3.38 |
| | Pooling | 12.01 | 12.20 | 7.55 | 11.40 |

performance degradation, regardless of design complexity. Notably, average pooling yields the best results among pruning approaches, likely because it partially preserves spatial structural features.

> ***Finding 1*** Advanced pruning methods provide limited benefits over simple baselines on VQA tasks, whereas all methods suffer systematic degradation on grounding tasks, with average pooling achieving the best performance.

## 2.2 UNVEILING TASK-DEPENDENT VISUAL PROCESSING PIPELINE

Building on the task-dependent performance degradation observed in Sec. 2.1, prior explainability studies on LLMs and VLMs (Selvaraju et al., 2016; Ding et al., 2017; Zhang et al., 2024; Yin et al., 2025) have not sufficiently explored how visual processing adapts to shifts in task focus, such as from VQA to VG. To address this, we conduct two analytical experiments: visualizing attention flows from the final token to vision tokens during decoding, and applying gradient-weighted attribution methods to trace critical visual information pathways across tasks. Additionally, we evaluate the model's object-centric perception at different stages using two fine-grained metrics.

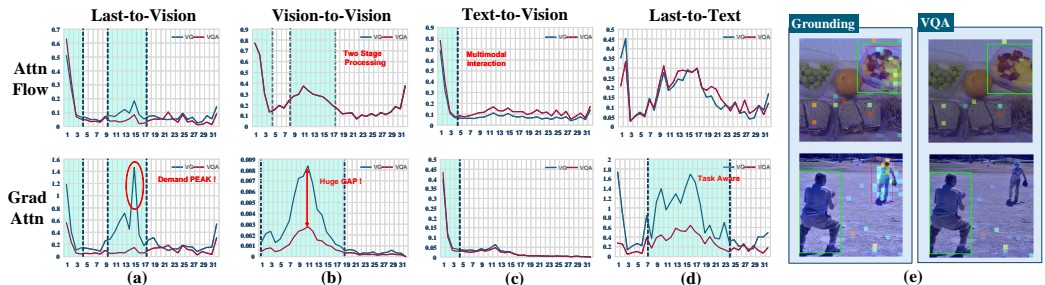

Figure 3: (a) to (d) show different types of attention flows (*First row*) and gradient-weighted attention flows (*Second row*), where A-to-B means the degree of attention A pays to B. (e) shows the differences in Last-to-Vision attention maps across different tasks. VLMs exhibit a two-stage visual processing pipeline, with task-independent multimodal interactions in early layers and task-specific processing in middle layers.

Figure 3 depicts task-dependent characteristics of visual processing in VLMs. Panels (a) and (b) at the first row show that attention flows exhibit distinct early and mid-stage phases. However, gradient-weighted analysis (Second row) reveals a pronounced task-dependent divergence in the mid-stage, underscoring the model's sensitivity to task requirements during visual integration — with VG tasks showing greater reliance on vision tokens. Panel (c) highlights a task-independent aspect: early multimodal interactions, suggesting universal visual processing in initial stages. Panel (d) illustrates task-varying differences in text information handling. Further experiments on attention blocking (Appendix B.5) indicate that, in VG tasks, textual cues extract critical visual details, resulting in unique last-to-text attention patterns.

**Visual Attention Entropy And Object-Centric Cohesion:** Attention flows offer insights into the model's information processing dynamics. To further quantify the multi-stage visual processing

pipeline identified in the prior analysis and its task-dependent characteristics, we introduce two fine-grained metrics: Visual Attention Entropy (VAE) and Object-Centric Cohesion (OCC). VAE measures the distribution of information in the visual self-attention mechanism by computing the average Shannon entropy across visual tokens (Eq. (1)). High VAE values indicate diffuse, global attention patterns, whereas low values reflect concentrated, local focus. Complementing this, OCC assesses object-level feature cohesion by calculating the Intersection over Union (IoU) between ground-truth object tokens and the top-$k$ tokens most similar to the object's center token (Eq. (2)). Higher OCC scores denote stronger localization of features to relevant objects, capturing fine-grained processing.

$$H(v_i) = -\sum_{j=1}^{i-1} p(v_j|v_i) \log_2 p(v_j|v_i), \quad \text{VAE} = \frac{1}{N-1} \sum_{i=2}^{N} H(v_i) \tag{1}$$

$$\text{OCC}(\mathcal{O}) = \frac{|V_k^{\text{model}} \cap V_{\mathcal{O}}|}{|V_k^{\text{model}} \cup V_{\mathcal{O}}|} \tag{2}$$

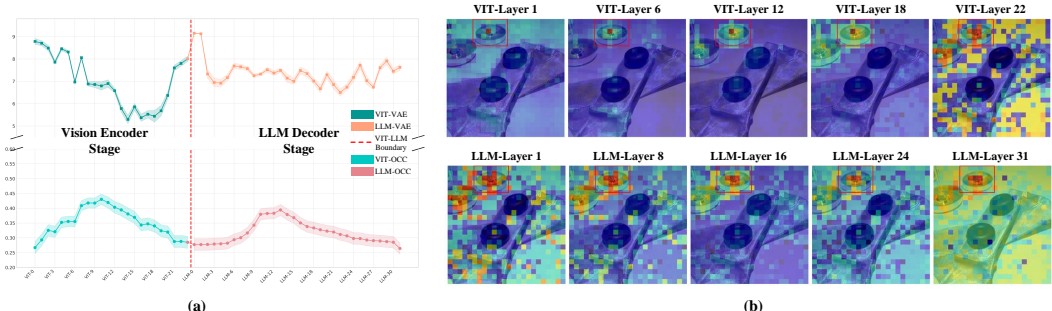

Figure 4: Visualization of VLM's Two-Stage Vision Tokens Processing: (a) Layer-wise Analysis of VAE and OCC Metrics; (b) Layer-wise Instance Heatmap Visualization. Both demonstrate fine-grained feature extraction at the mid-stage.

As shown in Figure 4, the VAE of the ViT encoder exhibits a decreasing trend in the middle stage, indicating a gradual shift from global context integration to fine-grained feature extraction. In contrast, the VAE of the LLM decoder fluctuates after a sharp initial increase, suggesting a more complex process of reorganizing visual features and integrating them into the textual semantic space. The OCC scores provide a clearer explanation — they peak in the middle stage of both ViT and LLM, signifying the formation of object-level representations. This phenomenon also effectively explains the earlier observation: why grounding tasks demand such high levels of visual information at this stage.

> ***Finding 2*** Visual processing in VLMs unfolds through a **multi-stage pipeline**, progressing from global semantic integration to fine-grained object-centric focus, with **task-specific** reliance on vision tokens. Grounding tasks require heightened visual integration during middle stages for spatial reasoning, in contrast to the reduced demands in image understanding tasks.

## 2.3 Spatial Integrity: Reconstructing the Global Reference Frame

Building on the mid-stage visual integration demands in Sec. 2.2, where pruning disrupts task-specific vision reliance, we hypothesize that **spatial integrity** — via the **Global Spatial Reference Frame** from position embeddings — is essential for spatial perception, as pooling methods' superior grounding performance indicates. To validate this, we design experiments restoring integrity through modified position embedding strategies.

### 2.3.1 A Taxonomy of Position Embedding Strategies

To rigorously test our hypothesis, we first deconstruct the implicit position embedding (PE) handling strategies within existing pruning methods, as shown in Figure 5, abstracting them into three distinct paradigms:

**Position Embedding Range Compression (PERC):** Compresses the PE of pruned tokens into a tiny range, missing the global reference frame, like Visionzip.

**Position Embedding Sparse Preservation (PESP):** Retains the original PE for each pruned token, forming a sparse subset within an incomplete spatial frame, like FastV.

**Relative Position Mapping Extension (RPME):** Preserves the relative spatial distance of the pruned tokens and extends their PE via linear mapping, to span the entire original range and retain the spatial integrity.

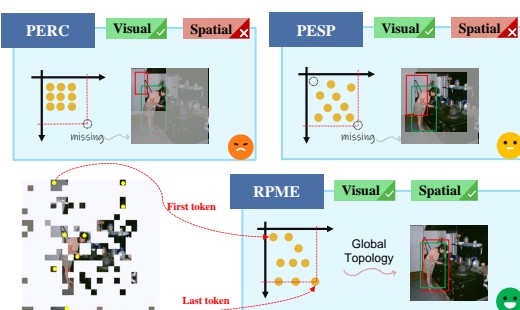

Figure 5: Sketch of different Position Embedding Strategies. RPME retains the spatial integrity.

**Experiment Setup** We select two representative methods, VisionZip (PERC) and FastV (PESP), replacing their PE strategy with RPME, and then evaluate the performance of these "fixed" models on visual grounding benchmarks.

Table 3: Position Reconstruction Experiment on Refcoco series and VQA Benchmarks. The symbols '+' and '-' indicate changes relative to the pre-reconstruction values showed in Table 2.

| Method | Refcoco -test | Refcoco -val | Refcoco+ -testA | Refcoco+ -testB | Refcoco+ -val | Refcocog -test | Refcocog -val | GQA | MMB | VQAv2 | MME |
|---|---|---|---|---|---|---|---|---|---|---|---|
| Vanilla | 58.30 | 56.42 | 59.43 | 38.88 | 46.32 | 48.50 | 48.82 | 61.9 | 64.7 | 78.5 | 1862 |
| **Average 64 Tokens** | | | | | | | | | | | |
| VisionZip-fix | 11.57 (+7.53) | 10.50 (+6.69) | 9.27 (+5.54) | 7.57 (+3.71) | 8.62 (+5.12) | 8.19 (+4.81) | 8.31 (+5.10) | 55.6 (+0.5) | 61.8 (+1.7) | 70.6 (-1.8) | 1700 (+10) |
| Fastv-fix | 4.52 (+1.79) | 4.11 (+2.10) | 3.84 (+2.67) | 2.75 (+1.73) | 4.31 (+1.90) | 4.17 (+1.98) | 4.22 (+2.21) | 46.2 (+0.1) | 47.8 (-0.2) | 54.1 (-0.9) | 1247 (-8) |
| Pooling | 12.01 | 11.84 | 12.20 | 7.55 | 10.50 | 11.40 | 9.85 | – | – | – | – |
| **Average 128 Tokens** | | | | | | | | | | | |
| VisionZip-fix | 21.39 (+16.90) | 21.04 (+16.93) | 19.96 (+15.90) | 13.45 (+8.59) | 16.10 (+12.22) | 15.69 (+12.19) | 15.52 (+12.04) | 58.5 (+0.9) | 63.4 (+1.4) | 74.3 (-1.3) | 1751 (-10) |
| Fastv-fix | 13.41 (+3.07) | 13.24 (+3.11) | 11.69 (+3.16) | 12.29 (+2.46) | 14.55 (+3.31) | 12.02 (+3.15) | 11.87 (+3.45) | 51.3 (+0.8) | 57.7 (+1.6) | 60.3 (-1.5) | 1494 (+4) |
| Pooling | 23.01 | 22.67 | 24.37 | 15.04 | 17.88 | 19.69 | 19.03 | – | – | – | – |

Results in Table 3 show that RPME yields notable improvements across benchmarks: VisionZip achieves gains of **5.6%** and **13.4%** in two settings, while FastV sees more modest increases of **1.8%** and **3.2%**. These differences confirm our analysis: PERC in VisionZip eliminates positional information, whereas PESP in FastV preserves absolute coordinates but disrupts spatial continuity. Gains grow with larger token budgets, underscoring the increasing importance of complete spatial frameworks for richer visual organization. Pooling methods outperform others consistently by aggregating features on coarse grids that implicitly maintain global topology, reinforcing that reconstructing continuous spatial coordinates is vital for grounding tasks. This strategy has a negligible impact on image understanding and reasoning benchmarks, indicating broad applicability.

> ***Finding 3*** The degradation of VLMs on grounding tasks is principally driven by the loss of Global Spatial Reference Frame within token pruning strategies, which can be restored by preserving global position embedding.

## 3 METHODOLOGY

Our analysis reveals that existing token pruning methods fail on spatial localization tasks by disrupting the global spatial reference frame. This motivates three core design principles for effective visual token compression: (1) preserving spatial uniformity to ensure consistent coverage; (2) aggregating redundant information in a vision-centric, cohesive manner while retaining local salience (Stage1); and (3) applying text-modulated fine-grained filtering to select task-relevant tokens based on textual semantics (Stage2). We apply these principles in Nüwa, a two-stage pruning framework.

### 3.1 STAGE 1: SPATIAL COHESION PRUNING IN THE VISION ENCODER

This stage reduces the initial $N^2$ visual tokens in the vision encoder to a dense, spatial-preserving sequence via three sequential operations.

### 3.1.1 SEPARATION VIA GRID PARTITIONING

To maintain spatial integrity, we partition the input token grid $\mathcal{T} = \{t_1, t_2, \ldots, t_{N^2}\}$ into $M \times M$ non-overlapping local regions $\mathcal{R}_{i,j}$. Subsequent selection and aggregation occur at the region level, enabling a complete global coordinate system.

### 3.1.2 ALIGNMENT VIA SALIENCE IDENTIFICATION

Within each region $\mathcal{R}$, we select representative **benchmark tokens** as aggregation centers. These tokens should exhibit high global salience; we initially use attention scores from the [CLS] token. However, analysis indicates sparse distributions in deeper vision encoder layers. To mitigate this, we incorporate information capacity, defined as the L2-norm of the token's **key vector** ($\|\mathbf{k}_i\|_2$), as a secondary criterion. The resulting **salience score** $S(t_i)$ for token $t_i$ is the product of its global attention score and information capacity:

$$S(t_i) = \alpha_{\text{cls},i} \cdot |\mathbf{k}_i|_2 \tag{3}$$

where $\alpha_{\text{cls},i}$ is the attention weight from the [CLS] token. In each local region $\mathcal{R}_k$, we select the $k$ tokens with the highest salience scores to form the **Benchmark Token set** $\mathcal{T}_B$.

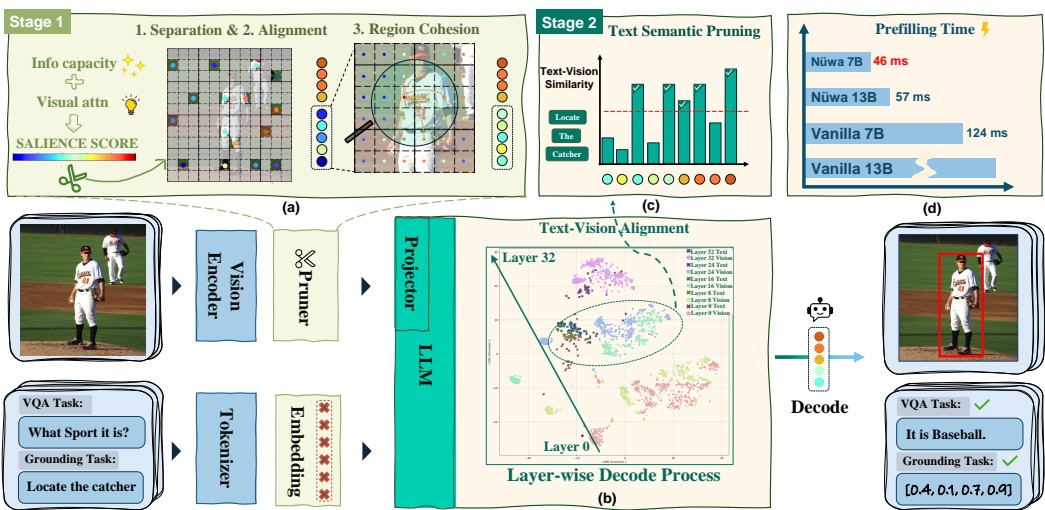

Figure 6: The Framework of Nüwa: (a) Stage 1 Pruning regarding Separation, Alignment and Cohesion; (b) Layer-wise 2D visualization of text-visual token similarity during LLM; (c) Stage2 pruning based on text semantics at LLM mid-stage; (e) prefill time of Nüwa across different scales.

### 3.1.3 AGGREGATION VIA SPATIAL PROXIMITY

This operation merges features from other tokens into the benchmark set $\mathcal{T}_B$, guided by role assignment and spatial proximity, yielding a semantically rich and spatially complete token sequence.

**Role Assignment: Pillars and Collectors.** We differentiate benchmark tokens in $\mathcal{T}_B$ by information capacity. Recent works (Darcet et al., 2024; Lappe & Giese, 2025) identify high-norm tokens in ViTs as **registers** — frequently attended during decoding and often task-agnostic. Modifications to these can shift feature distributions and affect predictions. Thus, we classify tokens with $\|\mathbf{k}_i\|_2$ in the top quartile as **Pillar Tokens** ($\mathcal{T}_P$), whose features remain unmodified. The rest are **Collector Tokens** ($\mathcal{T}_C$), which aggregate from spatial neighbors.

$$\mathcal{T}_P = \{t_i \in \mathcal{T}_B \mid \|\mathbf{k}_i\|_2 \geq \text{Quantile}(\{\|\mathbf{k}_j\|_2\}_{t_j \in \mathcal{T}_B}, 0.75)\}; \quad \mathcal{T}_C = \mathcal{T}_B \setminus \mathcal{T}_P \tag{4}$$

**Weighted Aggregation.** High semantic similarity does not imply aggregability; relying solely on it for global features is inadequate, as it risks disrupting object-centric representations by merging

spatially distant tokens. Thus, we balance it with spatial proximity to form a weight matrix $\mathbf{W} \in \mathbb{R}^{K \times N^2}$, where $K = |\mathcal{T}_B|$, combining semantic and proximity matrices.

**Semantic Similarity Matrix (A):** We consider only positively correlated semantic information. Element $A_{ij}$ is defined as Eq. (5):

$$A_{ij} = \text{ReLU}\left(\text{sim}(\mathbf{v}_i, \mathbf{v}_j)\right) = \text{ReLU}\left(\frac{\mathbf{v}_i \cdot \mathbf{v}_j}{|\mathbf{v}_i||\mathbf{v}_j|}\right) \tag{5}$$

**Spatial Proximity Matrix (P):** To penalize long range aggregation, we define a proximity matrix allowing each benchmark token to aggregate features within an extended local neighborhood, enabling limited cross-region interaction. Element $P_{ij}$ is computed as Eq. (6):

$$P_{ij} = 1 - \max\left(1, \frac{d(p_i, p_j)}{d_{\text{thresh}}}\right) \tag{6}$$

where $d(p_i, p_j)$ is the Euclidean distance between $p_i$ and $p_j$, and $d_{\text{thresh}}$ is a predefined threshold.

Based on role assignment, the final aggregation weight $W_{ij}$ is defined as:

$$W_{ij} = \begin{cases} \delta_{ij} & \text{if } t_i \in \mathcal{T}_P \text{ (Pillar Token)} \\ A_{ij} \cdot P_{ij} & \text{if } t_i \in \mathcal{T}_C \text{ (Collector Token)} \end{cases} \tag{7}$$

where $\delta_{ij}$ is the Kronecker delta, ensuring Pillar Tokens only aggregate from themselves.

The weight $\hat{\mathbf{W}}$ is row-normalized from $\mathbf{W}$, the original feature matrix is $\mathbf{V} \in \mathbb{R}^{N^2 \times D}$. The updated feature matrix for benchmark tokens, $\mathbf{V}'_B \in \mathbb{R}^{K \times D}$, is computed as $\mathbf{V}'_B = \hat{\mathbf{W}}\mathbf{V}$.

## 3.2 STAGE 2: TEXT-MODULATED PRUNING IN THE LLM

Following Stage 1, the aggregated vision tokens $\mathbf{V}'_B$ are fed into the LLM for multimodal feature interaction. We apply a second round of task-oriented pruning at an intermediate layer, after initial multimodal alignment (Shukor & Cord, 2024), where textual and visual features converge in a shared space. To guide this pruning, we first derive a holistic textual query vector $\bar{\mathbf{q}}$ by average-pooling the embeddings $\{\mathbf{q}_1, \ldots, \mathbf{q}_K\}$ of text tokens:

$$\bar{\mathbf{q}} = \frac{1}{K} \sum_{k=1}^{K} \mathbf{q}_k \tag{8}$$

We calculate a relevance score $R_i$ for each visual token $t'_i$ (with updated feature vector $\mathbf{v}'_i$, the $i$-th token of $\mathbf{V}'_B$) by measuring its cosine similarity to the query vector in the shared embedding space:

$$R_i = \text{sim}(\text{proj}(\mathbf{v}'_i), \bar{\mathbf{q}}) = \frac{\text{proj}(\mathbf{v}'_i) \cdot \bar{\mathbf{q}}}{|\text{proj}(\mathbf{v}'_i)| \cdot |\bar{\mathbf{q}}|} \tag{9}$$

where $\text{proj}(\cdot)$ denotes the multimodal projection layer mapping visual features into the common text-vision embedding space. Finally, we retain only the top-$K_{\text{final}}$ visual tokens with the highest relevance scores $R_i$, passed to subsequent LLM layers for final reasoning and response generation.

## 4 EXPERIMENT

**Experimental Setup:** To validate the generality and effectiveness of our method, we conduct experiments on multiple VLMs and diverse benchmarks for image understanding and visual grounding tasks. The evaluated models are LLaVA-1.5, LLaVA-NeXT. We use 10 VQA benchmarks (e.g., GQA, TextVQA) and 3 VG benchmarks (RefCOCO, etc.). All experiments are run on NVIDIA A100-40G GPUs. Detailed configurations are in the Appendix B.2.

### 4.1 MAIN RESULT

**Performance on VQA Tasks:** We apply Nüwa during the inference stage of LLaVA-1.5-7B. More results in Table 5 demonstrate that Nüwa achieves optimal performance across nearly all benchmarks, with average performance further improving upon existing SOTA methods. On more VLM

models with different scales, such as LLaVA-NeXT-7B, Nüwa consistently demonstrates performance gains, establishing its strong generalizability. Results can be found in Appendix B.2. **Performance on Visual Grounding Tasks:** Visual grounding tasks are highly sensitive to spatial information in tokens, constituting a critical evaluation dimension for compression methods. On RefCOCO series visual grounding benchmarks, as shown in Table 6, our method substantially outperforms alternative approaches, achieving approximately 35% performance improvement over previous methods under 64 average tokens configuration. When retaining 192 tokens, our method maintains 79% of the original model's performance.

**Efficiency Analysis:** As shown in Table 4, we evaluate efficiency from two dimensions: theoretical computational complexity and actual prefill latency. Nüwa introduces negligible computational overhead, with TFLOPs increasing by only 0.01 and prefill stage latency increasing by 1 ms, compared with previous SOTA. Nüwa's design requires executing attention computation only once on tokens from the final layer of the vision encoder, enabling seamless FlashAttention compatibility through simple code modifications.

Table 4: Comparison of Model Efficiency. "main" and "metric" mean the standard Transformer pipeline and the additional computational load of pruning metric.

| Method | Avg Token | main (TFLOPs) | metric (MFLOPs) | Prefill-Time (ms) |
|---|---|---|---|---|
| Vanilla | 576 | 5.9730 | 0 | 124 |
| FastV | 64 | 0.8341 | **4.7185** | 92 ↓ 26% |
| SparseVLM | 64 | 0.8141 | 5.5050 | 104 ↓ 16% |
| VisionZip | 64 | **0.6461** | 8.9128 | **45** ↓ 63% |
| Nüwa | 64 | 0.6476 | 17.5636 | 46 ↓ 62% |

## 4.2 ABLATION STUDY

**Ablation on Spatial Proximity Threshold** To enable aggregation based on spatial proximity, we define local neighborhoods via a distance threshold $\tau$. Empirical evaluation (Table 7) shows that performance peaks at $\tau = 26\%$ of the maximum distance. Smaller values restrict aggregation scope, leading to suboptimal results, while larger values incorporate noise from distant regions, also degrading performance. These results confirm the effectiveness of localized aggregation in preserving spatial integrity. **Ablation on Key Components** Experimental results in Table 8 show that region partitioning is essential for grounding tasks, as it implements a more precise RPME strategy, but has negligible effects on VQA tasks. The L2-norm criterion positively enhances baseline token selection across all tasks, consistent with our analysis in Sec. 3.1.3. For two-stage pruning, gains over random pruning remain modest. Notably, combining random pruning with region partitioning substantially degrades performance, as the partitioning introduces potentially task-irrelevant tokens that random selection may retain.

Table 5: VQA performance comparison On LLava-1.5 7B. **Best** and second-best results are highlighted.

| Method | Source | GQA | MMB | MMMU | MME | VQAv2 | VQAtext | POPE | SQA | SEED | MMVet | avg |
|---|---|---|---|---|---|---|---|---|---|---|---|---|
| Vanilla | CVPR'24 | 61.9 | 64.7 | 36.3 | 1862 | 78.5 | 58.2 | 85.9 | 69.5 | 58.6 | 31.1 | 100% |
| *Average Token 192 ↓ 66.7%* | | | | | | | | | | | | |
| FastV | ECCV'24 | 52.7 | 61.2 | 34.3 | 1612 | 67.1 | 52.5 | 64.8 | 67.3 | 57.1 | 27.7 | 89.53% |
| PDrop | CVPR'25 | 57.1 | 63.2 | 34.1 | 1766 | 74.9 | 56.1 | 82.3 | **70.2** | 54.7 | 30.5 | 95.87% |
| SparseVLM | ICML'25 | 57.6 | 62.5 | 33.8 | 1721 | 75.6 | 56.1 | 83.6 | 69.1 | 55.8 | 31.5 | 96.11% |
| VisionZip | CVPR'25 | 59.3 | 63.0 | **36.6** | 1782 | **76.8** | 57.3 | 85.3 | 68.9 | 56.4 | **31.7** | 98.26% |
| **Nüwa** | - | **60.9** | **64.3** | 35.5 | **1834** | 75.9 | **57.4** | **86.4** | 68.2 | **59.7** | 30.5 | **98.80%** |
| *Average Token 128 ↓ 77.8%* | | | | | | | | | | | | |
| FastV | ECCV'24 | 49.6 | 56.1 | 34.9 | 1490 | 61.8 | 50.6 | 59.6 | 60.2 | 55.9 | 28.1 | 85.04% |
| PDrop | CVPR'25 | 56.0 | 61.1 | 34.2 | 1664 | 73.5 | 55.1 | 82.3 | **69.9** | 53.3 | 30.8 | 94.32% |
| SparseVLM | ICML'25 | 56.0 | 60.0 | 33.8 | 1696 | 73.8 | 54.9 | 80.5 | 67.1 | 53.4 | 30.0 | 93.36% |
| VisionZip | CVPR'25 | 57.6 | 62.0 | **37.9** | 1761 | **75.6** | 56.8 | 83.2 | 68.9 | 54.9 | **32.6** | 97.63% |
| PruMerge | ICCV'25 | 57.8 | 59.6 | 36.2 | 1712 | 74.7 | 54.3 | 81.5 | 67.6 | - | 30.4 | 95.06% |
| **Nüwa** | - | **60.2** | **63.4** | 35.8 | **1828** | 75.1 | **57.0** | **85.5** | 67.8 | **58.7** | 29.8 | **97.87%** |
| *Average Token 64 ↓ 88.9%* | | | | | | | | | | | | |
| FastV | ECCV'24 | 46.1 | 48.0 | 34.0 | 1256 | 55.0 | 47.8 | 59.6 | 51.1 | 51.9 | 25.8 | 79.36% |
| PDrop | CVPR'25 | 41.9 | 33.3 | 26.5 | 1092 | 57.3 | 45.9 | 55.9 | 69.2 | 40.0 | 24.9 | 71.56% |
| SparseVLM | ICML'25 | 53.8 | 60.1 | 35.44 | 1589 | 68.2 | 53.4 | 77.5 | **69.8** | 51.1 | 24.9 | 89.93% |
| VisionZip | CVPR'25 | 55.1 | 60.1 | 36.2 | 1690 | 72.4 | **55.5** | 77.0 | 69.0 | 52.2 | **31.7** | 93.99% |
| PruMerge | ICCV'25 | 55.4 | 59.6 | 35.8 | 1616 | 71.3 | 52.0 | 75.7 | 69.5 | - | 28.0 | 91.71% |
| **Nüwa** | - | **58.3** | **62.0** | **36.4** | **1706** | **72.8** | 54.9 | **83.0** | 67.5 | **56.44** | 28.2 | **94.91%** |

Table 6: Performance comparison on the RefCOCO series benchmark On LLava-1.5 7B. **Best** and second-best results are highlighted.

| Method | Source | Refcoco-test | Refcoco-val | Refcoco+-testA | Refcoco+-testB | Refcoco+-val | Refcocog-test | Refcocog-val | avg |
|---|---|---|---|---|---|---|---|---|---|
| Vanilla | CVPR'24 | 58.30 | 56.42 | 59.43 | 38.88 | 46.32 | 48.50 | 48.82 | 100% |
| | | | | **Average Tokens 192 ↓ 66.7%** | | | | | |
| FEATHER* | ICCV'25 | 27.7 | - | 24.7 | - | - | 27.2 | - | 48.38% |
| **Nüwa** | - | **47.91** | **46.12** | **43.18** | **31.86** | **37.68** | **37.64** | **37.90** | **79.29%** |
| | | | | **Average Tokens 128 ↓ 77.8%** | | | | | |
| Fastv | ECCV'24 | 10.34 | 10.13 | 8.53 | 9.83 | 8.16 | 8.87 | 9.10 | 18.55% |
| SparseVLM | ICML'25 | 6.27 | 6.17 | 5.79 | 4.22 | 9.85 | 6.35 | 6.47 | 12.84% |
| VisionZip | CVPR'25 | 4.49 | 4.11 | 4.06 | 4.86 | 3.88 | 3.50 | 3.48 | 8.1% |
| **Nüwa** | - | **45.09** | **43.69** | **42.63** | **28.98** | **35.32** | **36.59** | **36.00** | **75.20%** |
| | | | | **Average Tokens 64 ↓ 88.9%** | | | | | |
| Fastv | ECCV'24 | 2.73 | 2.01 | 1.17 | 1.02 | 2.41 | 2.19 | 2.01 | 3.81% |
| SparseVLM | ICML'25 | 1.04 | 1.01 | 0.96 | 1.28 | 0.96 | 0.61 | 0.66 | 1.88% |
| VisionZip | CVPR'25 | 4.04 | 3.81 | 3.73 | 3.86 | 3.50 | 3.38 | 3.21 | 7.28% |
| **Nüwa** | - | **29.43** | **28.60** | **28.22** | **17.47** | **22.22** | **21.81** | **21.42** | **47.19%** |

Table 7: Ablation Study On cohesion distance. The best-performing result in each column is **bolded**, and the second-best is underlined.

| Config | GQA | MMB | MME | Refcoco-test | Refcoco+-testA | Refcoco+-testB | Refcocog-test |
|---|---|---|---|---|---|---|---|
| dist18 | 0.5784 | 60.2852 | 1695.37 | 0.2783 | 0.2818 | 0.1655 | 0.2018 |
| dist148 | **0.5853** | 61.6838 | 1704.981 | 0.2922 | **0.2936** | 0.1730 | **0.2189** |
| dist280 | 0.5833 | 62.0275 | 1706.869 | **0.2943** | 0.2822 | **0.1747** | 0.2181 |
| dist412 | 0.5826 | **62.1134** | **1711.202** | 0.2879 | 0.2705 | 0.1698 | 0.2135 |
| dist544 | 0.5811 | 62.0275 | 1702.986 | 0.2834 | 0.2637 | 0.1651 | 0.2100 |
| dist676 | 0.5810 | 61.9416 | 1696.32 | 0.2801 | 0.2590 | 0.1636 | 0.2083 |
| dist808 | 0.5808 | 61.8557 | 1706.869 | 0.2769 | 0.2607 | 0.1632 | 0.2071 |
| dist940 | 0.5799 | 61.9416 | 1705.986 | 0.2774 | 0.2572 | 0.1622 | 0.2059 |
| dist1058 | 0.5799 | 61.7698 | 1704.486 | 0.2765 | 0.2560 | 0.1630 | 0.2054 |

Table 8: Ablation Study on each design. Include Pillar-token selecting, Stage2 Random Pruning and Region Separation.

| region | pillar token | random S2 | GQA | MMB | MME | Refcoco test | Refcoco+ testA | Refcoco+ testB | Refcocog test |
|---|---|---|---|---|---|---|---|---|---|
| ✗ | ✗ | ✗ | 58.84 | 58.18 | 1791 | 6.83 | 6.54 | 4.50 | 5.58 |
| ✗ | ✗ | ✔ | 57.07 | 56.43 | 1736 | 6.72 | 6.48 | 4.38 | 5.25 |
| ✗ | ✔ | ✗ | 59.62 | 62.98 | 1807 | 6.35 | 6.12 | 4.65 | 5.60 |
| ✗ | ✔ | ✔ | 58.43 | 61.71 | 1771 | 7.01 | 6.81 | 4.42 | 4.92 |
| ✔ | ✗ | ✗ | 57.94 | 56.68 | 1742 | 43.50 | 39.85 | 26.10 | 34.20 |
| ✔ | ✗ | ✔ | 57.35 | 56.10 | 1724 | 43.17 | 38.94 | 25.58 | 33.74 |
| ✔ | ✔ | ✗ | **60.18** | **63.40** | **1828** | **45.09** | **42.63** | **28.98** | **36.59** |
| ✔ | ✔ | ✔ | 59.03 | 62.14 | 1791 | 44.30 | 41.20 | 27.50 | 35.80 |

# 5 CONCLUSION

In this paper, we identify limitations in existing token pruning methods for visual grounding tasks and perform a systematic analysis of VLMs' multi-stage visual processing pipelines. Results reveal task-specific demands, where grounding relies on global spatial reference frames disrupted by pruning. To mitigate this, we propose Nüwa, a two-stage framework with Boids-inspired aggregation and text-guided refinement to preserve spatial integrity. Extensive experiments across 13 datasets and multiple VLMs show state-of-the-art performance on VQA (95% retention) and VG (47.2% retention), with 89% TFLOPs and 62% prefill reductions via 88.9% token pruning.

## REPRODUCIBILITY STATEMENT

To ensure the reproducibility of the results presented in this paper, we take the following steps. **For the experiment of Sec. 2.1**, we set up simple baselines for each category of methods to conduct comparative experiments. Implementation details are provided in the Appendix B.3. **For the analytical experiments in Sec. 2.2**, no additional configurations are applied. **For the Experiment of Sec 2.3**, we conduct position re-estimation experiments. The algorithm implementation for RPME is provided in the Appendix B.4. All experiments are based on LLAVA-1.5 7B, with the same environment configuration, dataset, and model weights claimed in Appendix B.1. **For the Main Experiment**, we provide the algorithm implementation of the complex Stage-1 pruning in the Appendix B.2. The Stage-2 implementation is analogous to FASTV.

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

# A    RELATED WORK

## A.1    EFFICIENT LARGE VISION-LANGUAGE MODELS

LLMs and VLMs face significant computational efficiency challenges, particularly with extended sequences. LLMs grapple with the growing key-value (KV) cache during autoregressive inference, leading to the development of token reduction strategies like StreamingLLM (Xiao et al., 2023) and H2O (Zhang et al., 2023). However, VLMs confront amplified complexity due to the quadratic growth of visual tokens with image resolution or video frames, making their computational costs prohibitive and necessitating modality-specific optimizations. Two main architectural approaches address these computational constraints. One involves architectural compression, where modules like Q-Former (InstructBLIP (Dai et al., 2023)), perceiver resampler (OpenFlamingo (Awadalla et al., 2023)), and Locality-enhanced Abstractor (Honeybee (Cha et al., 2023)) distill high-dimensional visual inputs into compact representations, reducing the sequence length processed by expensive attention mechanisms. The other pathway utilizes hardware-aware optimization strategies, such as FlashAttention (Dao et al., 2022; Dao, 2023), which optimize memory access patterns for accelerated self-attention computation without altering token quantities, achieving performance gains through algorithmic refinements and efficient resource utilization.

## A.2    TOKEN PRUNING IN LARGE VISION-LANGUAGE MODELS

A complementary approach to VLM efficiency focuses on reducing computational overhead through token sequence optimization. The quadratic computational complexity of Transformer attention mechanisms becomes particularly problematic when processing the extensive visual token sequences typical in VLMs. Consequently, vision token pruning has emerged as a critical research direction, which can be systematically categorized along multiple dimensions. Token reduction approaches can be classified based on their training requirements into training-free and training-based methods. Regarding implementation stages, these techniques operate across four primary phases: (1) visual encoder preprocessing, (2) LLM internal processing, (3) KV cache optimization, and (4) hybrid multi-stage approaches. Each pruning strategy involves two fundamental decisions: identifying which tokens to retain and aggregating useful features from discarded tokens.

**Token Pruning At Vision Encoder**    ToME (Bolya et al., 2023) establishes the foundation for training-free token merging at the visual encoder stage, demonstrating effective feature-based token consolidation that influences subsequent works, including VisionZip (Yang et al., 2024), DivPrune (Alvar et al., 2025), LLaVA-PruMerge (Shang et al., 2024), and so on (Tong et al., 2025; Liu et al., 2025a). These methods leverage visual feature similarity to merge redundant tokens before they enter the language model, thereby reducing the computational burden on downstream processing stages.

**Token Pruning Within LLM**    FastV (Chen et al., 2024) pioneers attention score-based token pruning within the LLM processing pipeline, establishing a training-free paradigm that guides later developments such as SparseVLM (Zhang et al., 2025b), PyramidDrop (Xing et al., 2024), FastVLM (Vasu et al., 2024), and so on (Liu et al., 2025a; Arif et al., 2024; Khaki et al., 2025). These approaches dynamically identify and remove less informative tokens based on attention patterns during inference, maintaining model performance while significantly reducing computational requirements.

**Multi-Stage Optimization Strategies**    Comprehensive efficiency improvements have been achieved through multi-stage approaches that simultaneously optimize visual encoding, LLM prefill, and KV cache management during decoding. Representative methods include MustDrop (Liu et al., 2024b), LightVLM (Hu et al., 2025), and GlobalCom$^2$ (Liu et al., 2025b), which coordinate token reduction across multiple pipeline stages to maximize computational savings while preserving model capabilities.

**Training-Based Methods**    While training-based approaches may exhibit reduced generalizability compared to training-free methods, they demonstrate superior performance preservation through pruning-aware optimization. Methods such as M$^3$ (Cai et al., 2024), ATP-LLaVA (Ye et al., 2024), Dynamic-LLaVA (Huang et al., 2024), TokenPacker (Li et al., 2024), and TwigVLM (Shao et al.,

2025b) achieve competitive or superior performance compared to their full-token baselines through specialized training procedures that adapt the model to operate effectively with reduced token sequences.

# B    DETAILED EXPERIMENT SETUP

In this section, we provide detailed experimental setups and algorithm implementations, along with supplementary experiments. These include the main experiment, position reconstruction experiments, and attention blocking experiments.

## B.1    IMPLEMENT DETAILS

To ensure the reproducibility of the results presented in this paper, we have provided reproducible explanations for the key experiments in the paper.

Table 9: Important packages in the Conda Environment.

| Name | Version |
|---|---|
| datasets | 4.0.0 |
| llava | 1.2.2.post1 |
| lmms-eval | 0.3.4 |
| qwen-vl-utils | 0.0.14 |
| sentencepiece | 0.2.0 |
| tokenizers | 0.21.4 |
| torch | 2.6.0 + cu124 |
| torchaudio | 2.6.0 + cu124 |
| torchvision | 0.21.0 + cu124 |
| transformers | 4.54.0.dev0 |

**Models**    The model weights used are sourced from the Hugging Face community, specifically as follows:

1. LLAVA-1.5 7B
2. LLAVA-Next 7B
3. QWEN-2.5 VL 7B

**Datesets**    The dataset used originates from the lmm-lab datasets.

## B.2    MAIN EXPERIMENT

**Nüwa's stage-1 pruning algorithm:**    To more clearly illustrate the proposed method, particularly the complex first-stage cropping, we provide pseudocode for the algorithm implementation here Algorithm 1.

**A detailed setup for different VLMs:**    Based on different models, we adjust Nüwa's framework configuration, which still revolves around a two-stage process. The configuration calculation for Average Token ($\bar{R}_{v,\mathrm{LLM}}$) is as follows:

$$\bar{R}_{v,\mathrm{LLM}} = \frac{1}{L_l} \sum_{i=L_v}^{L_v+L_l-1} r_v^{(i)} \tag{10}$$

where $L_l$ is the number of LLM layers and $r_v^{(i)}$ is the number of visual tokens entering each LLM layer for processing. Our setting is shown in Table 10.

**LLAVA-Next 7B**    Performance evaluation of our method on LLAVA-Next 7B, including VG task at Table 11 and VQA task at Table 12. Nüwa achieves state-of-the-art performance across multiple datasets.

---

**Algorithm 1** Nüwa Stage-1 Pruning

---

**Require:**
 Input images $X \in \mathbb{R}^{B \times C \times H \times W}$
 Vision tower model (use CLIP VIT-Large here)
 Penalty threshold percentile $\tau = 0.25$
 Region configuration: top-$n$ tokens per $g \times g$ region
 Target tokens per image $k$
**Ensure:**
 Aggregated tokens $T \in \mathbb{R}^{B \times k \times D}$ (selected and aggregated features)
 Benchmark indices $I \in \mathbb{R}^{B \times k}$ (sorted positions of selected tokens)
1:  $B \leftarrow |X|;\quad N \leftarrow H \times W;\quad g, n, k \leftarrow Prune_{cfg}$  ▷ Get setting based on configuration
2:  $H_g, W_g \leftarrow H/g, W/g$
3:  $H_s, A \leftarrow Visiontower(X)$      ▷ Output hidden states and attentions
4:  $H \leftarrow H_s[L]\quad A_L \leftarrow A[L]$  ▷ Select layer $L = -2$ hidden states, shape $B \times (1 + N) \times D$
5:  $M \leftarrow \sum A_L[:, :, 0, 1:]$    ▷ Metric map from CLS attention, shape $B \times N$
6:  $M_{2D} \leftarrow$ Reshape $M$ to 2D     ▷ based on grid separation $H_g \times W_g$
7:  **Step1: Region-based Candidate Selection:**
8:  Unfold $M_{2D}$ into regions of $g \times g$ patches  ▷ Shape $B \times H_g \times W_g \times (g \times g)$
9:  **for** each region (i,j) **do**
10:  $C.append(Top_k(M_{2D}[:, i, j, :]))$ ▷ Select top-$n$ indices of Local coordinates within region
11: **end for**
12: $S_C \leftarrow$ Gather scores for $C$ from $M$
13: Select top-$k$ from $S_C$: $I \leftarrow$ Benchmark indices    ▷ Sorted
14: **Step2: Spatial Proximity And Similarity Construction:**
15: Create $P \in \mathbb{R}^{H_g \times W_g}$  ▷ Patch grid Spatial Proximity Matrix based on distance threshold
16: $\bar{H} \leftarrow$ Average hidden states from mid-stage  ▷ Shape $B \times (1 + N) \times D$
17: $P_t \leftarrow \bar{H}[:, 1:, :]$      ▷ Patch tokens, normalized
18: $Sim \leftarrow P_t \cdot P_t^T$      ▷ Similarity matrix, $B \times N \times N$
19: $W \leftarrow$ ReLU$(Sim) \odot P$   ▷ Aggregation weights with distance penalty
20: **Step3: Select Pillar Token:**
21: $S_I \leftarrow$ Gather scores for $I$ from $L2_{norm}$
22: $I_{pillar} \leftarrow Quantile(S_I, \tau)$
23: $W \leftarrow SetValues(W, I_{pillar}, 0)$  ▷ for Pillar Tokens, set 0 value for aggregation Weight $W$
24: **Step4: Aggregation:**
25: $W_I \leftarrow$ Gather $W$ for benchmark indices $I$    ▷ Shape $B \times k \times N$
26: Normalize $W_I$      ▷ Sum to 1 per row, self-weight 1
27: $T \leftarrow W_I \cdot H[:, 1:, :]$     ▷ Aggregate to benchmark tokens
28: **return** $T, I$

---

Table 10: Nüwa two-stage pruning setting on different VLMs.

| | Stage 1 | Stage 2 | Stage 1 | Stage 2 | Stage 1 | Stage 2 |
|---|---|---|---|---|---|---|
| LLAVA-1.5 | Average Token 64 | | Average Token 128 | | Average Token 192 | |
| | 112 | 16 | 224 | 32 | 336 | 48 |
| LLAVA-Next | Average Token 160 (5.6%) | | Average Token 320 (11.1%) | | Average Token 640 (22.2%) | |
| | 9% | 1% | 16% | 2% | 32% | 4% |
| Qwen2.5-VL | Average Token 25% | | Average Token 50% | | Average Token 75% | |
| | 35% | 42% | 60% | 66% | 85% | 70% |

Table 11: Refcoco series Benchmarks performance comparison On LLaVA-Next 7B. **Best** and second-best results are highlighted.

| Method | Refcoco-test | Refcoco+-testA | Refcoco-testB | Refcocog-test | avg |
|---|---|---|---|---|---|
| Vanilla | **77.73** | **76.34** | **57.25** | **71.05** | **100%** |
| Average Tokens 160 ↓ 94.4% | | | | | |
| Nüwa | 18.67 | 15.44 | 10.45 | 17.04 | 21.62% |
| Average Tokens 320 ↓ 88.9% | | | | | |
| Nüwa | 38.67 | 33.86 | 25.36 | 31.49 | 45.68% |
| Average Tokens 640 ↓ 77.8% | | | | | |
| Nüwa | **68.17** | **66.91** | **49.54** | **59.72** | **86.48%** |

Table 12: VQA Benchmarks performance comparison On LLaVA-Next 7B. **Best** and second-best results are highlighted.

| Methods | GQA | MMB | MME | POPE | SQA | TextVQA | avg |
|---|---|---|---|---|---|---|---|
| Vanilla | 64.2 | 67.9 | 1846 | 86.4 | 73.2 | 61.3 | 100% |
| Average Tokens 160 ↓ 94.4% | | | | | | | |
| SparseVLM | 51.2 | 52.1 | 1542 | 72.7 | 67.5 | 46.4 | 79.80% |
| VisionZip | 55.5 | 60.1 | 1628 | 74.8 | **68.3** | 56.2 | 87.60% |
| **Nüwa** | **60.0** | **60.4** | **1684** | 83 | 67.5 | **56.3** | **92.29%** |
| Average Tokens 320 ↓ 88.9% | | | | | | | |
| SparseVLM | 57.7 | **63.2** | 1685 | 82.2 | 67.3 | 55.9 | 91.20% |
| VisionZip | 59.2 | 63.1 | 1702 | 82.1 | 67.3 | **58.9** | 93.00% |
| **Nüwa** | **62.3** | **63.2** | **1813** | **86.0** | **68.2** | 58.5 | **96.10%** |
| Average Tokens 640 ↓ 77.8% | | | | | | | |
| SparseVLM | 60.3 | 65.8 | 1773 | 84.2 | 67.7 | 57.8 | 95.30% |
| VisionZip | 61.3 | **66.2** | 1787 | 85.9 | 68.1 | **60.2** | 96.90% |
| **Nüwa** | **63.4** | 65.4 | **1879** | **87.2** | **68.6** | 59.5 | **98.10%** |

**QWEN-2.5-VL 7B**    Performance evaluation of our method on QWEN-2.5 VL 7B, including VG task at Table 13 and VQA task at Table 14. Nüwa achieves state-of-the-art performance across multiple datasets.

Table 13: Refcoco series Benchmarks performance comparison On QWEN-2.5 VL 7B. **Best** and second-best results are highlighted.

| Method | Refcoco-testA | Refcoco-testB | Refcoco+-testA | Refcoco+-testB | Refcocog-test | avg |
|---|---|---|---|---|---|---|
| Vanilla | **92.56** | **85.16** | **89.02** | **79.15** | **87.24** | **100%** |
| | | | Average Tokens 75% | | | |
| Nüwa | 91.76 | 84.37 | 87.98 | 77.18 | 86.87 | 98.8% |
| | | | Average Tokens 50% | | | |
| Nüwa | 90.04 | 82.85 | 86.74 | 72.65 | 85.49 | 96.4% |
| | | | Average Tokens 25% | | | |
| Nüwa | 80.71 | 72.83 | 73.57 | 62.4 | 73.96 | 83.8% |

Table 14: VQA Benchmarks performance comparison On QWEN-2.5 VL 7B. **Best** and second-best results are highlighted.

| Methods | GQA | POPE | SQAimg | MMB-en | MME | VQA_text | avg |
|---|---|---|---|---|---|---|---|
| Vanilla | **61.9** | **87.9** | **77.8** | **83.5** | **2347** | **82.2** | **100.0%** |
| | | | Average Tokens 75% | | | | |
| **Nüwa** | 60.41 | 87.52 | 77.98 | 83.13 | 2340 | 77.35 | 98.5% |
| | | | Average Tokens 50% | | | | |
| **Nüwa** | 59.93 | 87.46 | 78.82 | 83.02 | 2330 | 76.03 | 98.1% |
| | | | Average Tokens 25% | | | | |
| **Nüwa** | 58.4 | 87.06 | 78.58 | 82.47 | 2313 | 73.81 | 96.9% |

## B.3    SIMPLE BASELINE COMPARISON EXPERIMENT

To provide a more accurate comparison, we make minor modifications to the baseline settings for each category of methods. Specifically, we only perform simple replacements on the pruning part. For random, we use `torch.randprem` with a random seed set to `seed(44)`. For pooling, we employ adaptive pooling as shown in Algorithm 2 to accommodate dynamic token inputs and cropping rates. The pseudocode is as follows:

---

**Algorithm 2** Adaptive Token Pooling Compress

---

**Require:**
    Input visual token sequence $X \in \mathbb{R}^{N \times d}$
    Original grid dimensions $(H_{in}, W_{in})$ (such that $H_{in} \times W_{in} = N_{in}$)
    Token retention ratio $\rho \in [0, 1]$
**Ensure:**
    Pooled visual token sequence $V' \in \mathbb{R}^{N_{out} \times d}$
1:  $k_{target} = \lfloor N_{in} \times \rho \rfloor$                           ▷ Calculate target token
2:  $\alpha = H_{in}/W_{in}$
3:  $W_{out} = \text{round}(\sqrt{k_{target}/\alpha})$; $H_{out} = \text{round}(\alpha \times W_{out})$   ▷ Calculate New grid based original aspect ratio
4:  $N_{out} = H_{out} \times W_{out}$                       ▷ Final number of output tokens
5:  $F_{grid} = \text{Reshape}(X, (1, d, H_{in}, W_{in}))$          ▷ Reshape to 2D feature map
6:  $F_{pooled} = \text{Pooling}(F_{grid}, \text{output\_size} = (H_{out}, W_{out}))$ ▷ This operation automatically handles scaling and pooling
7:  $X' = \text{Flatten}(F_{pooled})$
8:  **return** $X'$

---

### B.4 POSITION RECONSTRUCTION EXPERIMENT

Here, we conduct a detailed exploration of the positional reconstruction experiment. For methods like Visionzip that employ PERC, the positional embedding is regenerated for the pruned complete sequence. Assuming ideal pruning, where the target is perfectly extracted, PERC effectively crops the image target and feeds it into an LLM for grounding. This approach naturally cannot output a bounding box based on the original image coordinate system. For PESP used in methods like FastV and SparseVLM, the original position embedding is employed, which performs relatively well but still compromises spatial integrity unless the target is located at the bottom-right corner of the image. This explains why such methods excel in grounding tasks. In contrast, RPME effectively scales the target to restore its original coordinate system, aiding localization of larger objects. However, it still fails with small objects, explaining why Visionzip's localization accuracy improves marginally after adopting RPME. Here, we provide the pseudocode for RPME:

---

**Algorithm 3** Relative Position Mapping Extension

---

**Require:**
    Pruned visual token Indices $I \in \mathbb{R}^{1 \times k}$ (sorted ascending, $k$ is the number of pruned tokens)
    Original Vision Position ID $(s, e)$ (Start and End indices of full visual range)
**Ensure:**
    New Position Embedding $V' \in \mathbb{R}^{k \times d}$ (remapped embeddings for pruned tokens)
1:   $max_I \leftarrow I[k-1]$                                             ▷ Last index as reference span
2:   $new\_pos \leftarrow [s]$                                          ▷ Anchor first pruned token to start
3: **for** $j = 1$ **to** $k - 1$ **do**
4:      $scaled \leftarrow ceil\left(I[j] \times \frac{e-s}{max_I}\right) + s$
5:      $new\_pos$.append$(scaled)$
6: **end for**
7: $V' \leftarrow$ get_position_embedding$(new\_pos)$       ▷ Retrieve embeddings for new positions; shape $k \times d$
8: **return** $V'$

---

### B.5 ATTENTION BLOCK EXPERIMENT

To better understand the multimodal information interaction process within VLMs, we conduct attention blocking experiments on three datasets, as shown in Table 15. We divid tokens into four main categories: (1) System tokens, (2) Visual tokens, (3) Text tokens, and (4) Last tokens, which represent the token that will be used to predict. The attention-blocking experiment is conducted based on LLAVA-1.5 7B. The model is divided into four equal phases according to the decoder layers, with attention blocking applied to each phase.

Table 15: Attention Blocking Experiments on LLAVA-1.5 7B.

| Blocked Layer | Vision to Vision | | | Text to Vision | | | Last to Vision | | |
|---|---|---|---|---|---|---|---|---|---|
| | GQA | MMB | Refcoco-test | GQA | MMB | Refcoco-test | GQA | MMB | Refcoco-test |
| Original | 61.9 | 64.7 | 58.30 | 61.9 | 64.7 | 58.30 | 61.9 | 64.7 | 58.30 |
| 0-7 | 60.5 | 61.62 | 18.17 | 39.38 | 51.63 | 54.75 | 62.55 | 64.26 | 57.85 |
| 8-15 | 55.2 | 57.98 | **2.64** | 53.32 | 46.39 | 14.11 | 62.47 | 64.08 | **2.01** |
| 16-23 | 58.44 | 62.37 | 19.27 | 61.04 | 62.45 | 15.94 | 61.32 | 64.26 | 64.52 |
| 24-31 | 58.81 | 62.37 | 19.31 | 61.12 | 63.05 | 15.08 | 62.12 | 64.17 | 59.10 |

#### B.5.1 BLOCKING ATTENTION FROM VISION TO VISION

In this setting, we block self-attention between visual tokens in each phase. This disrupts the model's ability to construct global and local visual contexts within the LLM decoder.

**Impact on QA Tasks (GQA, MMB):** Performance shows a certain degree of decline, but it is not catastrophic. For example, on GQA, even when blocking the most critical layers 8-15 (performance drops from 61.9 to 55.2), the model retains most of its performance. **Layer-wise Sensitivity:**

Disabling intermediate layers (8-15) causes the most significant performance degradation. This indicates that intermediate layers in the visual encoder are crucial for integrating low-level features extracted by earlier layers (e.g., edges, textures) into semantically meaningful object-level representations. Disabling early layers (0-7) or deep layers (16-31) has relatively minor impacts, likely because early features contain redundancy while deep features are highly abstracted.

**Impact on grounding tasks (Refcoco):** Performance crashes dramatically. Particularly when blocking layers 8-15, performance plummets from 58.30 to 2.64. **Layer sensitivity:** Similar to VQA tasks, intermediate layers (8-15) are absolutely critical, validating our previous experimental observations

**Summary:** In stark contrast to VQA tasks, localization tasks are extremely dependent on spatial and contextual relationships between visual features. This can be explained by our spatial integrity hypothesis and the global coordinate reference system. For instance, to understand "the cup to the left of the table", the model must first establish spatial adjacency between "table" and "cup" via vision-to-vision attention. Blocking vision-to-vision attention reduces the image to a collection of disjointed, unconnected visual patches, preventing the model from forming a coherent understanding of the scene structure — thus causing the grounding task to fail completely.

### B.5.2 BLOCKING ATTENTION FROM LAST TO VISION

In this setting, we block cross-attention from last tokens to vision tokens in each phase. This directly prevents the model from extracting visual information during decoding.

**Impact on VQA tasks (GQA, MMB):** Minimal impact on VQA Tasks. GQA performance varies slightly from 61.9 to 62.55 (within error margins, with a marginal improvement), and MMB decreased marginally from 64.7 to 64.26.

**Impact on grounding tasks (Refcoco):** Catastrophic failure on VG tasks, particularly when blocking layers 8-15, where performance drops to 2.01. Moreover, blocking attention in subsequent stages actually improves accuracy.

**Summary:** The results demonstrate that answer generation in VQA tasks depends minimally on vision, as disconnecting visual inputs at any stage has a negligible impact on performance. VQA tasks primarily leverage global semantic information extracted from early multimodal interactions. In contrast, the result of grounding tasks contrasts with initial expectations. Integrating Text-to-Vision attention and gradient-weighted flows from Sec. 2.2 (last-to-text), we hypothesize that visual grounding tasks rely more on spatially aware visual information processed in the model's mid-stage. The process then continually leverages textual cues to extract features, enabling integration of complex visual semantics within the visual modality (as analyzed in Sec. 2.2). Furthermore, blocking the Last-to-Vision attention in subsequent stages actually improves accuracy. We attribute this to the fact that subsequent visual information requires higher-level semantic integration to be transferred to the output space, which disrupts the spatial information in later stages.

### B.5.3 BLOCKING ATTENTION FROM TEXT TO VISION

In this setting, we block cross-attention from text tokens to vision tokens in each phase. This blocks multimodal information extraction.

**Impact on VQA tasks (GQA, MMB):** Performance degrades substantially, especially when blocking early layers (0-7), with GQA dropping from 61.9 to 39.38. Recovery occurs gradually as the blocked layer depth increases.

**Impact on grounding tasks (Refcoco):** Performance suffers severely across all blocked layers, particularly in mid-to-late stages (8-15, 16-23). This intriguing result highlights that grounding requires sustained multimodal interactions, yet early ones have minimal influence — consistent with anomalies in text-to-vision attention observed in Sec. 2.2.

**Summary:** These findings indicate that VQA tasks rely on text extracting global abstract features from visual inputs during initial processing for comprehension, with subsequent response generation depending little on vision. In contrast, visual grounding tasks exhibit a different reliance: they depend more on ongoing visual feature extraction in later stages, particularly when spatial understanding emerges, rather than early features, which lack enough spatial information.

> ***Finding 4*** Attention blocking experiments further reveal that grounding tasks primarily rely on two sets of information: 1. Abstract semantic information continuously extracted from text tokens to visual tokens; 2. Precise positional information was extracted from the model's mid-stage. Meanwhile, most VQA tasks predominantly depend on abstract semantic information extracted from text tokens to visual tokens during the model's early to mid-stages.

## B.6 BENCHMARKS

We utilize several benchmarks that evaluate a model's ability to understand static images, including their content, context, and associated textual queries.

**GQA (Hudson & Manning, 2019)**   The GQA benchmark is structured around three core components: scene graphs, questions, and images. It enriches visual content with comprehensive spatial information and object-level attributes. Questions are specifically designed to evaluate models' ability to comprehend visual scenes and reason about diverse aspects of images.

**MMBench (Liu et al., 2023)**   The MMBench Benchmark evaluates models through three hierarchical levels of abilities. The first level (L-1) focuses on fundamental perception and reasoning capabilities. The second level (L-2) expands into six distinct sub-abilities, while the third level (L-3) further refines these into 20 specific dimensions. This structure enables a granular and comprehensive assessment of a model's various capabilities. Additionally, MMB-CN is the Chinese version of the benchmark.

**MME (Fu et al., 2023)**   MME comprehensively evaluates models' perceptual and cognitive abilities across 14 subtasks. By employing manually constructed instruction-answer pairs and concise instructions, it effectively minimizes data leakage and ensures fair assessment of model performance.

**MMMU (Yue et al., 2023)**   MMMU evaluates multimodal models on complex tasks requiring college-level knowledge and reasoning. It includes 11.5K curated questions from exams, quizzes, and textbooks, spanning six disciplines: Art & Design, Business, Science, Health & Medicine, Humanities & Social Science, and Tech & Engineering. Covering 30 image types like charts, diagrams, and chemical structures, MMMU challenges models with advanced perception and domain-specific reasoning.

**MMVet (Yu et al., 2023)**   MMVet defines six core vision-and-language (VL) capabilities: recognition, OCR, knowledge, language generation, spatial awareness, and math. These capabilities integrate to address a range of complex multimodal tasks. MMVet evaluates 16 specific integrations of these capabilities through quantitative assessments.

**POPE (Li et al., 2023b)**   The POPE benchmark systematically evaluates object hallucination in models through a series of binary questions about object presence in images. Using accuracy, recall, precision, and F1 score as metrics, it precisely measures hallucination levels across different sampling strategies.

**ScienceQA (Lu et al., 2022)**   ScienceQA encompasses a wide array of domains, including natural, language, and social sciences, with questions hierarchically organized into 26 topics, 127 categories, and 379 skills. Through this comprehensive structure, it offers diverse scientific questions that effectively evaluate multimodal understanding, multi-step reasoning capabilities, and interpretability.

**SEEDBench (Li et al., 2023a)**    SEEDBench comprises 19,000 human-annotated multiple-choice questions evaluating models across 12 distinct aspects. It comprehensively assesses capabilities in recognizing spatial and temporal patterns within both images and videos.

**TextVQA (Singh et al., 2019)**    TextVQA focuses on the integration of textual information within images. It evaluates a model's ability to interpret visual elements and embedded text in images through tasks, requiring both visual and textual comprehension to answer questions accurately.

**VQA-V2 (Goyal et al., 2016)**    VQA-V2 evaluates models' visual perception capabilities through open-ended questions about 265,016 real-world scene images. Each question contains 10 human-annotated ground truth answers, enabling thorough assessment of a model's ability to interpret and respond to visual queries.

These benchmarks are specifically designed to evaluate a model's ability to locate textual descriptions of specific objects or regions within an image.

**RefCOCO, RefCOCO+, RefCOCOg (Yu et al., 2016)**    These datasets are standard benchmarks for evaluating referring expression comprehension.

- **RefCOCO** and **RefCOCO+** provide a large number of images annotated with referring expressions for objects. RefCOCO+ expands on RefCOCO by increasing the diversity of expressions and objects.
- **RefCOCOg** is an extension that includes more natural, grammatically complex, and longer referring expressions, challenging models to understand more nuanced linguistic descriptions and their grounding in complex scenes. These datasets typically evaluate models on their ability to accurately locate the target object described by the expression.

## C    DETAILED FLOPS CALCULATION

We provide a detailed derivation of the Floating Point Operations (FLOPs) for a VLM with a flexible, layer-wise token pruning mechanism.

### C.1    PRELIMINARIES AND NOTATIONS

We first define the key hyperparameters of the underlying LLM, which follows a standard Transformer architecture. These parameters, summarized in Table 16, form the basis of our calculations.

Table 16: Model architecture parameters used for FLOPs calculation. Example of LLAVA-1.5 7B.

| Symbol | Value | Description |
|---|---|---|
| $\mathcal{L}$ | 32 | Total number of Transformer layers |
| $H$ | 4096 | Hidden dimension of the model |
| $I$ | 11008 | Intermediate dimension of the Feed-Forward Network (FFN) |
| $S^{(l)}$ | Variable | Sequence length (number of tokens) at layer $l$ |

Our calculation assumes that one FLOP corresponds to one multiply-accumulate (MAC) operation, which involves two floating-point operations (a multiplication and an addition). This is a standard convention in analyzing the computational cost of neural networks.

### C.2    FLOPS OF A STANDARD TRANSFORMER LAYER

The computational cost of a single Transformer layer is dominated by two components: the Multi-Head Self-Attention (MHA) block and the Feed-Forward Network (FFN). We formulate the FLOPs for each, given a sequence of length $S$.

**Multi-Head Self-Attention (MHA).** The MHA mechanism involves four primary matrix multiplication steps:

1. **Q, K, V Projection:** Projecting the input sequence of shape $(S, H)$ into Query, Key, and Value matrices. This involves three separate weight matrices of size $(H, H)$.

$$\text{FLOPs}_{\text{QKV}} = 3 \times (2 \cdot S \cdot H \cdot H) = 6SH^2$$

2. **Attention Score Calculation:** Computing the dot product between Query and Key matrices, $(S, H) \times (H, S)$.

$$\text{FLOPs}_{\text{Scores}} = 2 \cdot S^2 \cdot H$$

3. **Value Aggregation:** Multiplying the attention scores (after softmax) with the Value matrix, $(S, S) \times (S, H)$.

$$\text{FLOPs}_{\text{Values}} = 2 \cdot S^2 \cdot H$$

4. **Output Projection:** Projecting the aggregated value matrix back to the hidden dimension via a weight matrix of size $(H, H)$.

$$\text{FLOPs}_{\text{Proj}} = 2 \cdot S \cdot H \cdot H = 2SH^2$$

The total FLOPs for one MHA block is the sum of these components:

$$\text{FLOPs}_{\text{attn}}(S) = 8SH^2 + 4S^2H \tag{11}$$

**Feed-Forward Network (FFN).** The FFN block consists of two linear transformations with a non-linear activation in between.

1. **Up-Projection:** Expanding the hidden dimension from $H$ to $I$.

$$\text{FLOPs}_{\text{up}} = 2 \cdot S \cdot H \cdot I$$

2. **Down-Projection:** Reducing the dimension from $I$ back to $H$.

$$\text{FLOPs}_{\text{down}} = 2 \cdot S \cdot I \cdot H$$

The total FLOPs for one FFN block is:

$$\text{FLOPs}_{\text{ffn}}(S) = 4SHI \tag{12}$$

Thus, the total computational cost for a single Transformer layer is:

$$\text{FLOPs}_{\text{layer}}(S) = \text{FLOPs}_{\text{attn}}(S) + \text{FLOPs}_{\text{ffn}}(S) \tag{13}$$

### C.3 Generalizing to Dynamic Token Pruning

Token pruning methods dynamically alter the sequence length $S$ as data propagates through the model's layers. A given pruning strategy can be formally described by a **pruning schedule**, which specifies the sequence length at each layer. Let $S^{(l)}$ denote the number of tokens present at the input of layer $l$, where $l \in \{0, 1, \ldots, \mathcal{L} - 1\}$.

For a baseline model without pruning, $S^{(l)}$ is constant for all layers. For a model with pruning, $S^{(l)}$ becomes a piece-wise constant function that is non-increasing with $l$. For instance, a method that prunes tokens before layer $l_p$ would have $S^{(l)} = S_{\text{initial}}$ for $l < l_p$ and $S^{(l)} = S_{\text{pruned}}$ for $l \geq l_p$.

To calculate the total FLOPs for the entire LLM under a specific pruning schedule, we sum the FLOPs of each layer using its corresponding input sequence length. The total computational cost is given by:

$$\text{FLOPs}_{\text{Total}} = \sum_{l=0}^{\mathcal{L}-1} \text{FLOPs}_{\text{layer}}(S^{(l)}) \tag{14}$$

This generalized Eq. (14) provides a robust framework to accurately quantify the computational savings of any token pruning method, regardless of where or how many times pruning is applied. Our analysis primarily accounts for matrix multiplications, which dominate the computational cost in large Transformers. The cost of other operations, such as LayerNorm, activation functions, and softmax, is considered negligible, consistent with prior work in the field (e.g., Kaplan et al., 2020). By applying this methodology, we ensure a fair and insightful comparison across all evaluated methods in our experiments.

### C.4 COMPUTATIONAL OVERHEAD OF PRUNING METRICS

A comprehensive analysis of token pruning methods must account for the computational overhead incurred by the pruning decision logic itself. While pruning reduces the overall FLOPs of the main computational graph, the process of selecting which tokens to prune introduces an additional, non-negligible cost. This overhead is critical for a holistic comparison, as a method with a computationally expensive metric might offset the gains from pruning. In this subsection, we formalize the FLOPs required for various pruning metrics.

First, we define the computational cost for common metrics used by the methods under evaluation. Let $S_v$ be the number of vision tokens and $S_q$ be the number of query tokens (e.g., a CLS or text token, where $S_q = 1$).

- **Attention Score-based Metric:** This involves computing the dot product between $S_q$ query vectors and $S_v$ key vectors, both of dimension $H$. The cost is:

$$\text{FLOPs}_{\text{attn\_score}}(S_q, S_v) = 2 \cdot S_q \cdot S_v \cdot H \tag{15}$$

- **Cosine Similarity Metric:** This metric requires computing the pairwise cosine similarity between $S_v$ vision tokens. The dominant cost is the dot product for each of the $\frac{S_v(S_v-1)}{2}$ pairs. The cost is approximated by:

$$\text{FLOPs}_{\text{cosine}}(S_v) = (S_v)^2 \cdot H \tag{16}$$

- **L2-Norm Metric:** Calculating the L2-norm for $S_v$ vectors of dimension $H$ requires approximately:

$$\text{FLOPs}_{\text{norm}}(S_v) = 2 \cdot S_v \cdot H \tag{17}$$

Using these formulations, we summarize the total computational overhead for each pruning method in Table 17. The total FLOPs of a method is the sum of the main computation FLOPs (Eq. 14) and the overhead FLOPs detailed here.

Table 17: Summary of Computational Overhead for Different Token Pruning Methods. $S_{v,i}$ denotes the number of vision tokens at the input of pruning stage $i$, and $S'_{v,i}$ denotes the number of remaining vision tokens after stage $i$. The initial number of vision tokens is 576.

| Method | Pruning Stage | Metric Used & Token Count | Overhead FLOPs Formula |
|---|---|---|---|
| FSATV | Stage 1 | Last token attention to vision tokens ($S_{v,1} = 576$) | $\text{FLOPs}_{\text{attn\_score}}(1, 576)$ |
| Pdrop & SparseVLM | Stage 1 | Text token attention to vision tokens ($S_{v,1} = 576 \rightarrow S'_{v,1} = 66$) | $\text{FLOPs}_{\text{attn\_score}}(1, 576)$ |
| | Stage 2 | Text token attention to vision tokens ($S_{v,2} = 66 \rightarrow S'_{v,2} = 30$) | $\text{FLOPs}_{\text{attn\_score}}(1, 66)$ |
| | Stage 3 | Text token attention to vision tokens ($S_{v,3} = 30 \rightarrow S'_{v,3} = 17$) | $\text{FLOPs}_{\text{attn\_score}}(1, 30)$ |
| VisionZip | Stage 1 | CLS token attention to vision tokens ($S_{v,1} = 576 \rightarrow S'_{v,1} = 64$) | $\text{FLOPs}_{\text{attn\_score}}(1, 576)$ |
| | Stage 2 | Cosine similarity among remaining vision tokens ($S'_{v,1} = 64$) | $+\text{FLOPs}_{\text{cosine}}(64)$ |
| **Nüwa** | Stage 1 | CLS token attention to vision tokens ($S_{v,1} = 576 \rightarrow S'_{v,1} = 112$) | $\text{FLOPs}_{\text{attn\_score}}(1, 576)$ |
| | | Cosine similarity & L2-norm among remaining tokens ($S'_{v,1} = 112$) | $+2 * \text{FLOPs}_{\text{cosine}}(112)$ |
| | Stage 2 | Text-Vision Cosine similarity ($S_v = 112$) | $\text{FLOPs\_cosine}(112+N\_\text{text})$ |

## D VISUALIZE RESULTS

In this section, we supplement with additional visualization results, including some case studies, VAE and OCC visualizations from visual encoders (CLIP and SIGLIP2) and more visual representations of the conclusions in the paper.

### D.1 LLAVA

**Attention flow** Here, we present the complete attention flow, encompassing all token types.

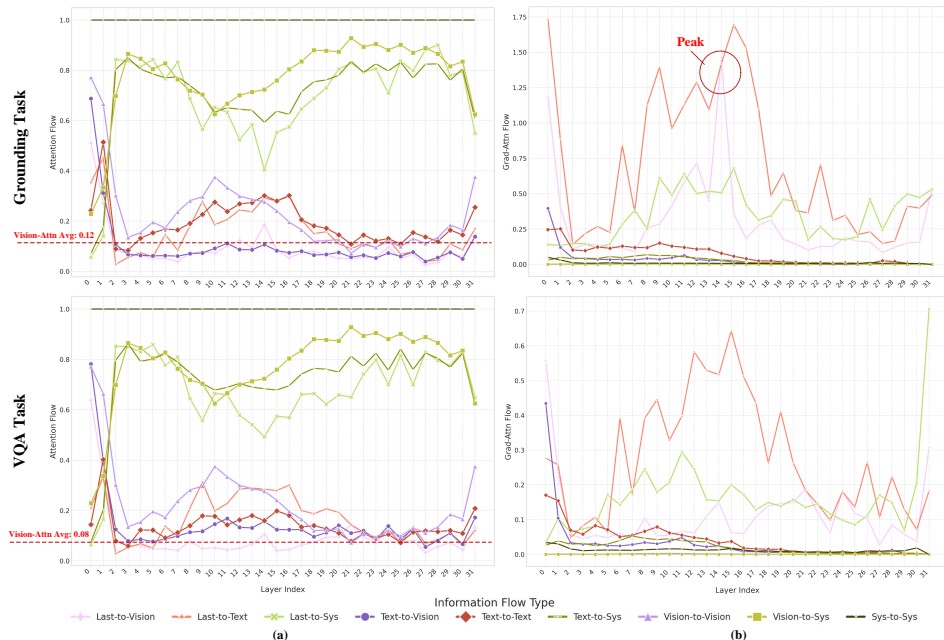

Figure 7: Complete Attention Flow for VQA and VG Tasks on LLAVA-1.5 7B.

**Text-Vsion Alignment** Figures 8 and 9 illustrate the layer-by-layer alignment process of vision and text tokens within the LLM. Complementing this, we present the layer-wise similarity heatmap, which together demonstrate the alignment status of multimodal data across different stages.

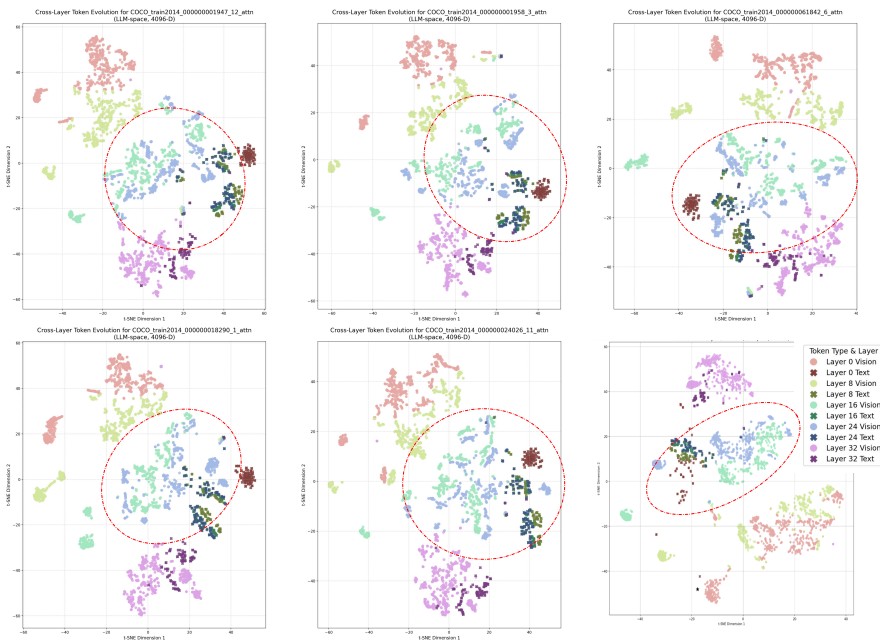

Figure 8: Two-Dimensional Visualization of Vision-Text Similarity.

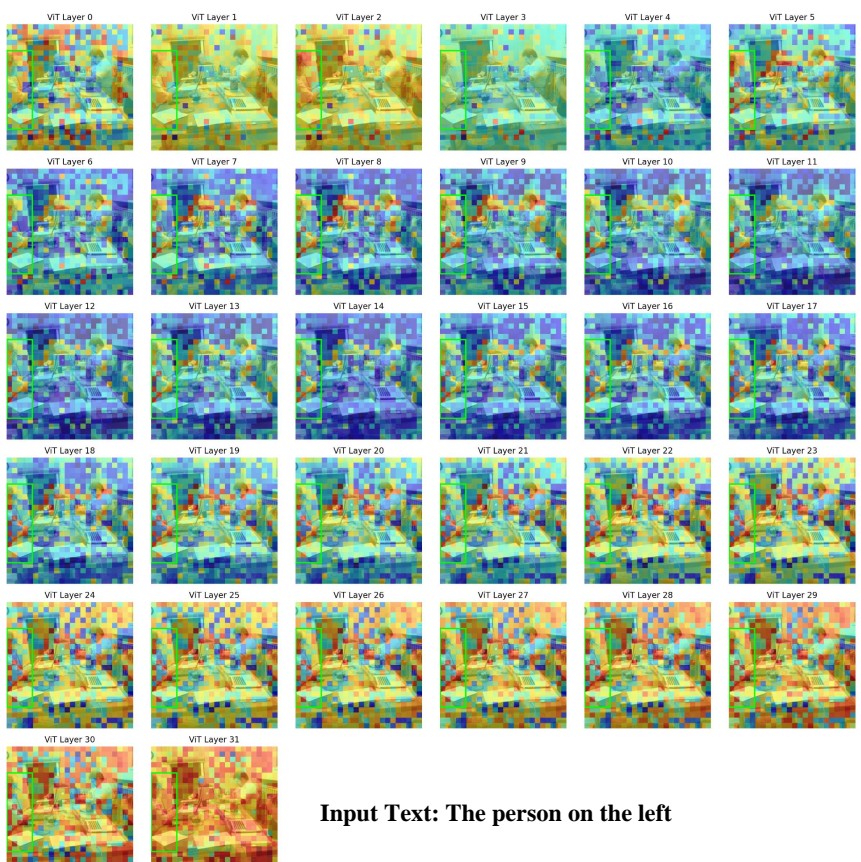

Figure 9: Visualization results of Layer-wise Vision-Text Similarity Heatmap.

**Pillar Token** We present three types of heatmaps in the Figure 10. The second row displays a heatmap with token L2 norm values, with the top and bottom rows showing tokens that made positive contributions during decoding for the VQA task and grounding task, respectively.

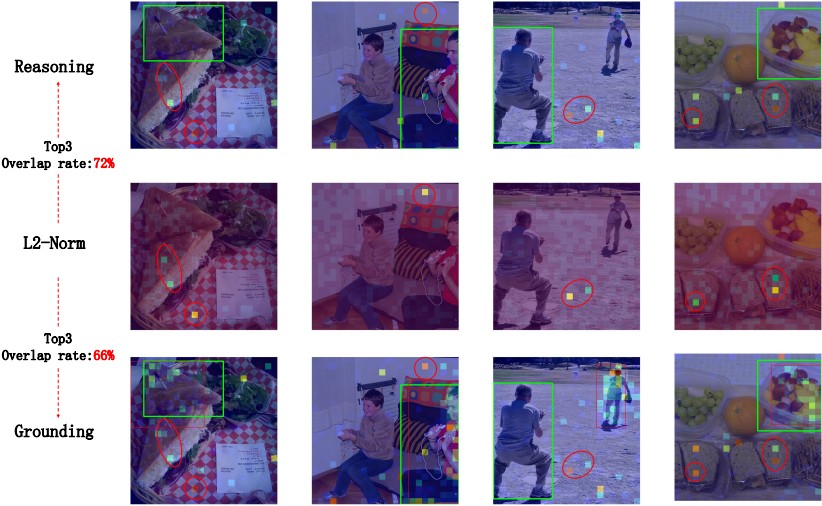

Figure 10: Visualization of "register" token and making significant contributions to the prediction (gradient-weighted attention values).

Here, we conduct further visual analysis on the "register" token. As mentioned in Section 3.1.3, these high-norm tokens receive significant attention in both VQA and grounding tasks. We calculate the overlap rate between the top-3 tokens with the highest L2 norm in the "register" token and the top-3 tokens with the highest weighted attention scores in the VQA and VG tasks, finding a high overlap rate of 72% in VQA and 66% in VG.

**Pruning Results**  We present some cropped visualization results Figure 11. Our approach preserves the integrity of the global space.

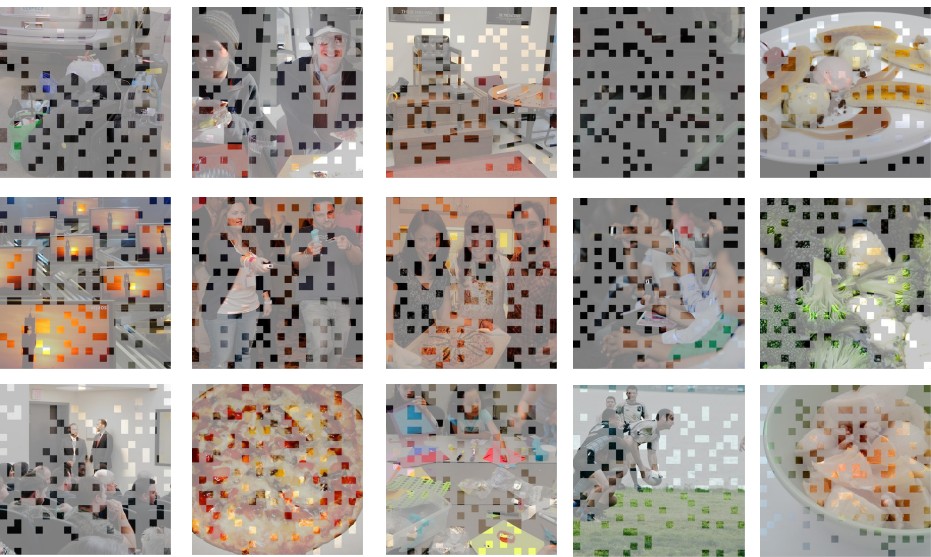

Figure 11: Visualization results of Pruning results.

## D.2 CLIP

We provide additional visualizations of attention maps and object-centric similarity maps during the CLIP processing (Figure 12).

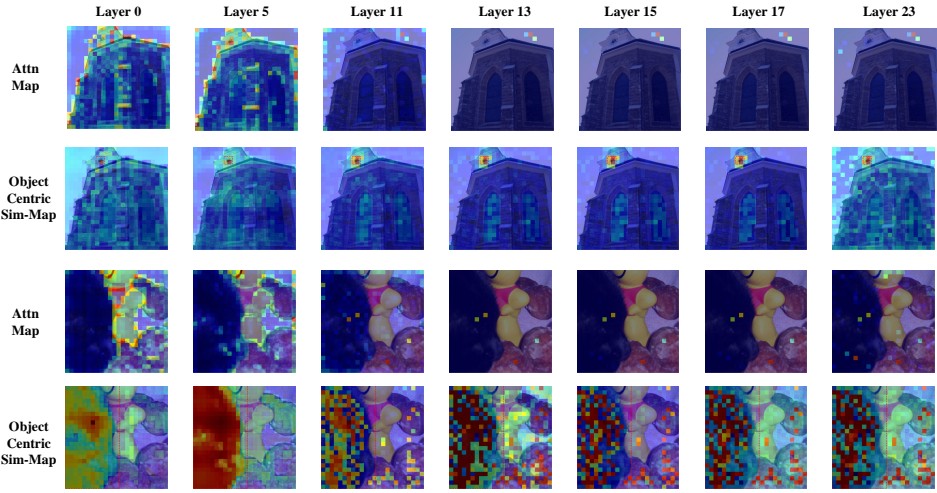

Figure 12: More visualization results processed by CLIP-VIT.

### D.3 SIGLIP

Since SIGLIP lacks a CLS token, the attention map visualization is based on the attention values received by each token. We provide some sample visualization results (Figure 14) and the metric — VAE and OCC (Figure 13). By examining visual results and metrics such as OOC, it can be observed that compared to CLIP, SIGLIP exhibits a later phase of fine-grained feature extraction, and its overall trend is less pronounced than that of CLIP.

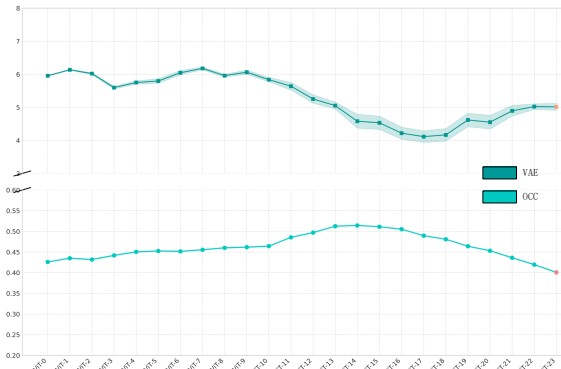

Figure 13: The VAE and OCC metric form SIGLIP2.

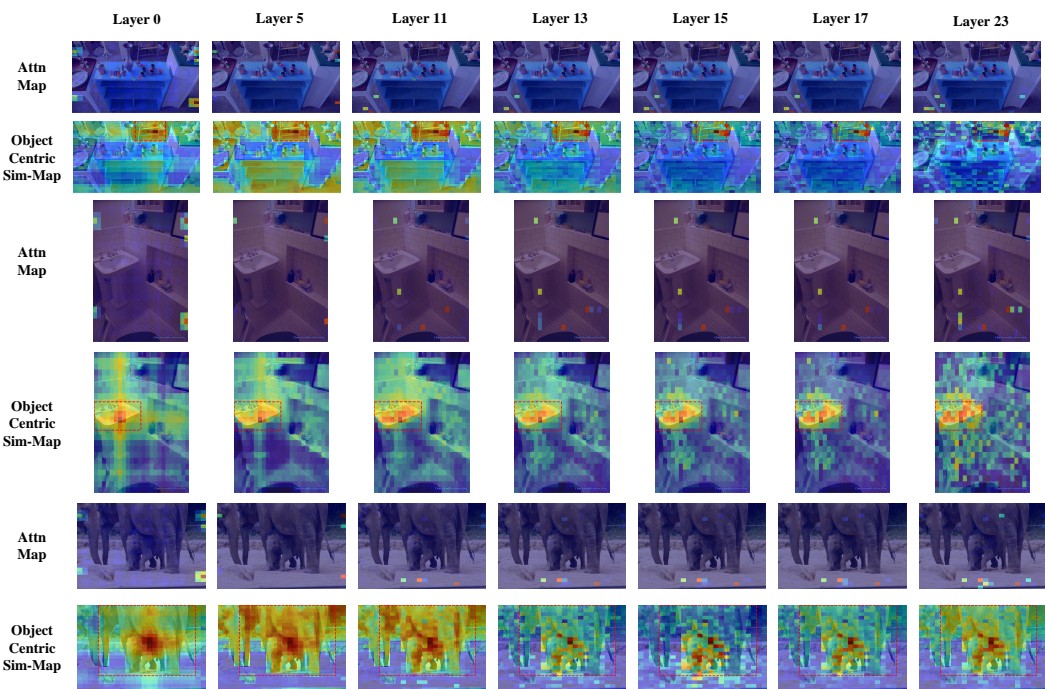

Figure 14: More visualization results processed by SIGLIP2.

## E  CASE STUDY

In this subsection, we present the results of vision token pruning and localization under several configurations of the Nüwa framework. Specifically, we evaluate three primary settings: 1) token selection without regional partitioning, 2) employment of Relative Position Mapping Extension (RPME),

and 3) the Nüwa with full configuration. Furthermore, we analyze a failure case wherein localization inaccuracies arise from the model's inherent misinterpretation of the text and the designated task.

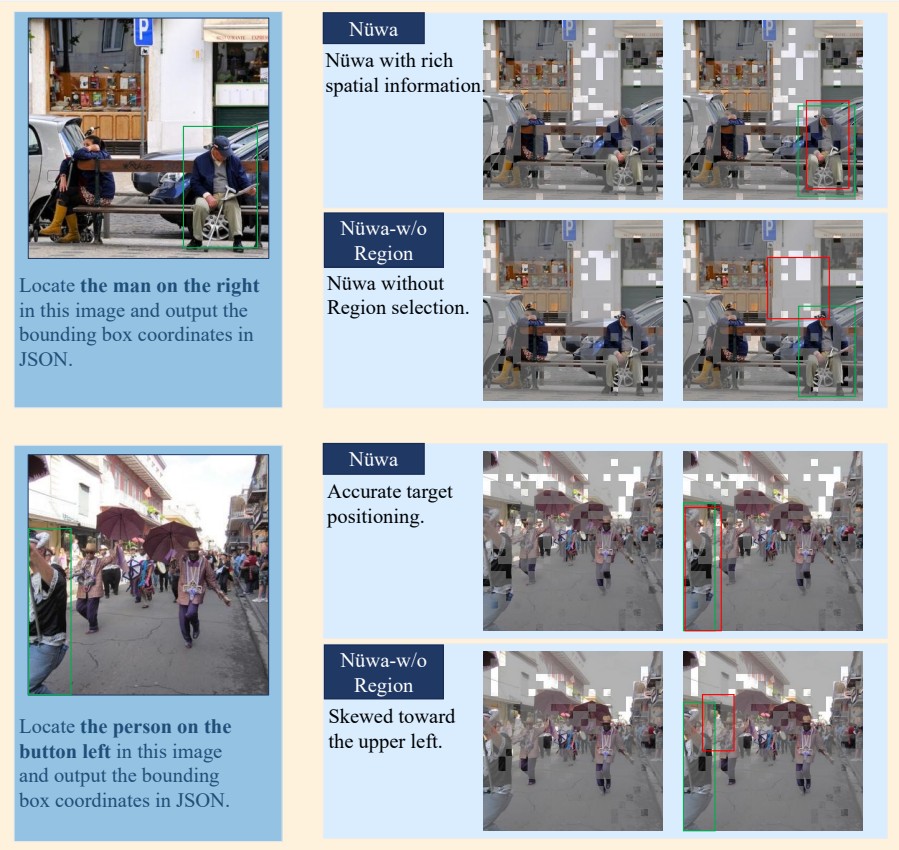

Figure 15: A comparison between the original Nüwa setting and the token selection setting without regional partitioning. Red and green bounding boxes represent the prediction and ground truth, respectively.

**Regional partitioning provides a significant advantage** As illustrated in Figure 15, the configuration employing regional partitioning demonstrates superior preservation of spatial integrity during token pruning, enabling the observation of all four corners of the image. Conversely, the failure to retain sufficient spatial information appears to impair the model's perception of global positioning, resulting in predictions that are systematically biased towards the top-left corner. The underlying reasons for this phenomenon are discussed in Section 2.3.

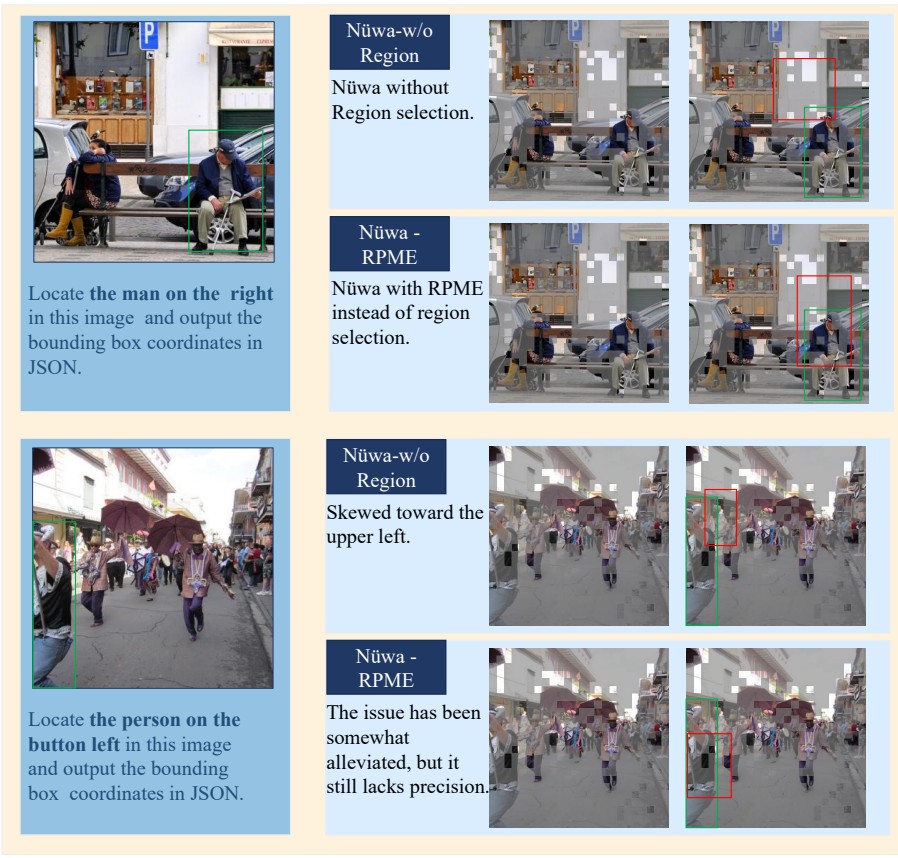

Figure 16: A comparison between the token selection setting without regional partitioning and the setting with RPME. Red and green bounding boxes represent the prediction and ground truth, respectively.

**RPME can mitigate the loss of spatial information.** As shown in Figure 16, the prediction results from the RPME-enabled setting are more accurate. Specifically, the localization no longer exhibits a consistent bias towards the top-left corner. However, while the overall position is correct, the shape (i.e., aspect ratio) of the predicted box is problematic. This may be attributed to the image 'distortion' introduced by the stretching operations in RPME, which could interfere with the model's understanding of the object's geometry.

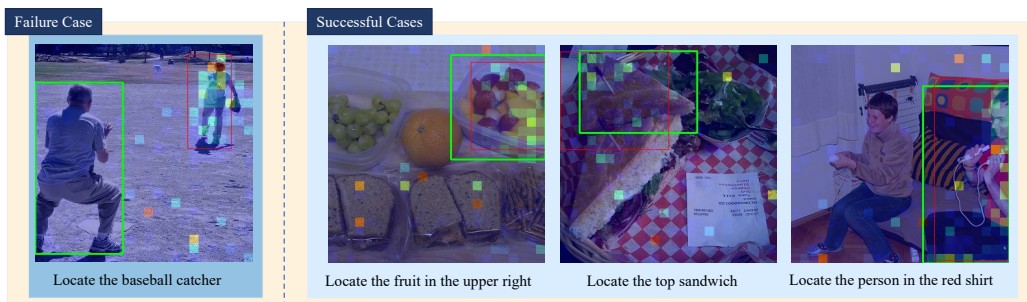

Figure 17: Localization failure attributable to the model's comprehension. Red and green bounding boxes represent the prediction and ground truth, respectively.

**Localization Failures Attributable to Model comprehension.**    As shown in Figure 17, another potential failure case for localization is presented, which arises from the VLM's incorrect understanding of the target. In the successful case on the right, we observe a clear positive correlation between the model's predicted bounding box and the regions where it assigns high attention to the vision tokens. In the failure case on the left, however, the model mistakenly identifies the person in the distance wearing a baseball glove as the target, assigns extremely high attention to that region, and consequently produces an incorrect localization.

