# OpenReview forum: "Nüwa: Mending the Spatial Integrity Torn by VLM Token Pruning"
_ICLR.cc/2026/Conference — ICLR 2026 Poster_

### Official Review · Reviewer_MQfG · 2025-10-25

**Soundness:** 3
**Presentation:** 3
**Contribution:** 2
**Rating:** 4
**Confidence:** 4

**Summary:**

The paper studies why vision token pruning in vision-language models hurts visual grounding much more than VQA, and proposes a new pruning framework called Nüwa to fix it. The authors analyze the VLM visual processing pipeline and argue that grounding depends on a global spatial reference frame built from positional embeddings and mid-layer vision tokens, while most pruning methods destroy this frame. They present a two-stage pruning method: (1) before the LLM, partition vision tokens into regions, pick salient tokens, and aggregate nearby tokens into them while keeping global spatial anchors intact; (2) inside the LLM mid-layers, prune again using text–vision similarity to keep only task-relevant visual tokens. They claim large speedups while keeping VQA performance and sharply improving grounding retention.

**Strengths:**

1. The analysis of attention/gradient flow, visual attention entropy, and object-centric cohesion across vision and LLM layers is thoughtful. It supports the claim that grounding relies on mid-stage spatial structure, not just final semantic alignment. This is valuable diagnostic science for the community.
2. Insight about position embeddings. The taxonomy of PE handling and the position reconstruction experiment is convincing.
3. Method makes sense relative to the diagnosis. Stage 1 is explicitly designed to preserve a dense global spatial scaffold. This is more principled than naive semantic clustering. Stage 2 then does task-aware pruning in LLM space, which matches the observed task divergence in mid layers. The story is coherent end to end.
4. They compare to random and pooling. They acknowledge that pooling is surprisingly strong and then explain why (implicit topology preservation).

**Weaknesses:**

1. The idea of a two-stage token pruning pipeline is not new,  similar observations and motivations have appeared in prior and concurrent works [1-9]. The authors should more carefully compare against these approaches, and make explicit what is actually novel here.
2. The RefCOCO results reported for the baselines differ notably from those in other works for example [2]. I may be misunderstanding the experimental settings, so the authors should clarify their setup and explain the discrepancy.
3. No failure analysis, for example, whether the text-conditioned Stage 2 introduces text bias (the model may discard visual evidence that contradicts misleading text).

[1] PyramidDrop: Accelerating Your Large Vision-Language Models via Pyramid Visual Redundancy Reduction.
[2] VScan: Rethinking Visual Token Reduction for Efficient Large Vision-Language Models.
[3] GLDS: Global–Local Diversity Selection for Scalable Token Pruning in Vision–Language Models.
[4] Token Pruning in Multimodal Large Language Models: Are We Solving the Right Problem?
[5] Feather the Throttle: Revisiting Visual Token Pruning for Vision-Language Model Acceleration.
[6] Boosting Multimodal Large Language Models with Visual Tokens Withdrawal for Rapid Inference.
[7] ST3: Accelerating Multimodal Large Language Model by Spatial-Temporal Visual Token Trimming.
[8] Cross-modal Information Flow in Multimodal Large Language Models.
[9] LOOK-M: Look-Once Optimization in KV Cache for Efficient Multimodal Long-Context Inference.

**Questions:**

1. How does this paper differ from the prior two-stage pruning approaches, and what are the novel contributions?
2. The RefCOCO results reported for the baselines differ notably from those in other works. Could the authors clarify their experimental settings and explain the reason for this discrepancy?
3. Has any failure analysis been conducted?

---

> ### Author Response · Authors · 2025-11-20
> **Response 1**
>
> **Q1: How does this paper differ from the prior two-stage pruning approaches, and what are the novel contributions?**
> ## R1:  Clarifying Contributions and Methodological Comparisons
>
> We thank the reviewer for the detailed summary of related work. The core technical contributions of our work are primarily twofold:
> - 1. **In-depth Analysis and Insights:** We conduct a systematic experimental analysis of why the widespread performance degradation of existing VLM pruning methods occurs on **Visual Grounding (VG) tasks**, and we explain the root causes of this issue.
> - 2. **Targeted Framework Design:** Based on the insights from our analysis, we propose a two-stage framework (Nüwa) with several novel modules specifically designed to solve this localization performance degradation.
>
> To further clarify the specific innovations of our work, we will compare it against each of the cited works in the order you have listed:
>
> - **Compare with PyramidDrop [1]:**
>
> PyramidDrop prunes tokens at multiple layers within the LLM. This design repeatedly intervenes in the feature flow, disrupting the original spatial relationships that are critical for precise localization. In contrast, **our method's region-based pruning, performed outside the LLM**, preserves the macroscopic spatial reference from the outset. Furthermore, our second-stage pruning is a single, more precise intervention at layer 16, the optimal point for multimodal feature alignment.
> - **Compare with VScan [2]:**
>
> Our contribution lies not only in **technical differences** (e.g., we use a "Pillar-Aggregate" token division instead of "local + global" tokens; our selection metric combines attention with information capacity), but more importantly, in a fundamental difference in research motivation. Our work is explicitly driven by the goal of **analyzing and solving the performance collapse on VG tasks**, whereas VScan does not focus on this as its core problem or contribution.
> - **Compare with  GLDS [3]:**
>
> The GLDS method, which is a token pruning work based on the Determinantal Point Process (DPP) diversity mechanism, has significant differences compared to our approaches, such as token selection and aggregation. As we verified, this is **concurrent work** under review.
> - **Compare with  Paper [4]:**
>
> This work served as an inspiration for our research, acting as a "problem-poser" by first highlighting issues like the VG performance collapse. Our work takes the next step as a "problem-solver" by conducting **a deeper analysis** of the causes of this phenomenon and proposing a **more comprehensive and effective solution**.
> - **Compare with  Feather the Throttle [5] :**
>
> This work attributes the localization problem to RoPE position encoding. Our analysis leads to a more general conclusion: the root cause is the loss of global contextual information. Therefore, our technical approach (region-aware pruning) is fundamentally different from theirs (global uniform sampling + modified attention) in both its starting point and design.
> - **Compare with  Paper [6]:**
>
> This work proposes dropping all visual tokens in the later layers of the LLM. Through a fair comparison using the equivalent average token count, their method requires ~288 tokens to achieve performance comparable to ours at ~192 tokens. This demonstrates that our method has **a clear advantage in computational efficiency.**
> - **Compare with  ST3 [7]:**
>
> **Similar to PyramidDrop, ST3 prunes tokens layer-by-layer within the LLM**, which also risks disrupting spatial information. Additionally, its focus includes KV Cache pruning, and its experimental setup is inconsistent with current mainstream methods, making a direct quantitative comparison difficult.
> - **Compare with  Paper [8]:**
>
> This is a VLM interpretability paper. However, our analysis is more in-depth; for example, we introduce gradient-weighting to differentiate token importance across tasks and propose new instance-level analysis metrics (VAE and OCC), which directly inform our algorithm design.
> - **Compare with Paper [9]:**
>
> **This paper is from the field of embodied AI**, with a method focused on **KV Cache pruning* for multi-turn robotic action generation. Its task setting, model architecture, and datasets are all different from the VLM domain, making it not directly comparable.
>
> **In summary, our contributions are:**
> - 1. We provide a systematic analysis of new insights into the performance degradation of VLM pruning on localization tasks.
> - 2. Based on this analysis, we propose a solution with several novel components that effectively address this problem.

---

> ### Author Response · Authors · 2025-11-20
> **Response 2**
>
> **Q2: The RefCOCO results reported for the baselines differ notably from those in other works. Could the authors clarify their experimental settings and explain the reason for this discrepancy?**
> ## R2: Regrading the RefCOCO results
>
> We sincerely thank the reviewer for this valuable feedback. We clarify that our reported baseline results are consistent with related works, and the apparent discrepancy stems from **the different base Vision-Language Models (VLMs) used for evaluation.**
>
> Specifically, our experiments are conducted on** LLaVA-1.5 and LLaVA-Next**, whereas the results in the paper that you mentioned are benchmarked on **Qwen2.5-VL**. It is expected that different base models will report different performance metrics on the same benchmark due to their varying capabilities.
>
> To provide a more intuitive comparison, we benchmark our results against **a recent work [5]** that utilizes the same LLaVA base model. As shown in the table below, the results reported in [5] are **largely consistent with our reproduction (Page 9; Table 6) of the LLaVA model.**
>
> ### Main Experiment on paper [5] (Page 7; Table 1)
>
> -  φ_original：means the FSATV
>
> **Setting:  pruning layer = 3 with reduction rate=75%**
>
> | Category | Criteria | FLOPS Red | Avg | OCID-Ref | RefCOCOg | RefCOCO+ | RefCOCO |
> | :--- | :--- | :--- | :--- | :--- | :--- | :--- | :--- |
> | **Attention-based** | φ_original | 68% | 5.9 | 5.7 | 5.1 | 6.1 | 6.7 |
> | | φ_R (Ours) | 68% | **16.7** | **22.9** | **15.1** | **13.3** | **15.3** |
> | | Δ | | +10.7 | +17.2 | +10.0 | +7.3 | +8.6 |
> | **Non-attention-based** | φ_KNN | 66% | 23.9 | 15.1 | 24.9 | 26.0 | 29.6 |
> | | φ_uniform | 66% | **28.0** | **20.6** | **28.6** | **29.7** | **33.3** |
> | **Ensemble** | φ_R + φ_uniform (Ours) | 61% | **27.2** | **29.1** | **27.2** | **24.7** | **27.7** |
>
>
> ### More Experiment including VisionZip on paper [5](Page 12; Table A3)
>
> | Method | FLOPs Red | Avg | OCID-Ref | RefCOCOg | RefCOCO+ | RefCOCO |
> | :--- | :--- | :--- | :--- | :--- | :--- | :--- |
> | **Baseline** | 0% | 53.2 | 40.7 | 56.3 | 55.0 | 60.9 |
> | **Baseline (pos shuffled)** | 0% | 8.0 | 9.0 | 7.8 | 7.1 | 8.0 |
> | **FasterVLM** | 65% | 5.7 | 8.0 | 5.9 | 4.2 | 4.7 |
> | **VisionZip** | 65% | 8.5 | 7.3 | 9.0 | 8.1 | 9.5 |
> | **FEATHER** | 64% | **39.3** | **33.1** | **40.1** | **39.7** | **44.1** |
> | **FEATHER (pos shuffled)** | 64% | 5.3 | 5.3 | 4.8 | 5.2 | 5.8 |
>
>
> Furthermore, to fully address your concern and to demonstrate the generalizability of our method, we have also **conducted a full set of experiments on Qwen2.5-VL (Page 18; Table 13, Table 14).** The results are as follows:
>
> | Retain Token| Method | refcoco-testA | refcoco-testB | refcoco+-testA | refcoco+-testB | refcocog-test | Average |
> | :--- | :--- | :--- | :--- | :--- | :--- | :--- | :--- |
> | | **Vanilla** | 92.56 | 85.16 | 89.02 | 79.15 | 87.24 | 100.00% |
> | **75%** | **Nüwa**  |91.76 | 84.37 | 87.98 | 77.18 | 86.87 | 98.8% |
> | **50%** | **Nüwa** |90.04 | 82.85 | 86.74 | 72.65 | 85.49 | 96.4% |
> | **25%** |**Nüwa** | 80.71 | 72.83 | 73.57 | 62.4 | 73.96 | 83.8% |
>
>
> | Retain Token|  Method | GOA | POPE | SQAimg | MMB-en | MME | VQA_text | Average |
> | :--- | :--- | :--- | :--- | :--- | :--- | :--- | :--- |:--- |
> | | **Vanilla** | 61.9 | 87.9 | 77.8 | 83.5 | 2347 | 82.2 | 100.0% |
> | 75%|  **Nüwa** | 60.41 | 87.52 | 77.98 | 83.13 | 2340 | 77.35 | 98.5% |
> | 50%|  **Nüwa** | 59.93 | 87.46 | 78.82 | 83.02 | 2330 | 76.03 | 98.1% |
> | 25%|  **Nüwa** | 58.4 | 87.06 | 78.58 | 82.47 | 2313 | 73.81 | 96.9% |
>
> **Further Analysis on Performance Discrepancy between Qwen2.5-VL and LLaVA:**
> During our experiments, we also observe that Qwen2.5-VL exhibits greater robustness to token pruning compared to LLaVA. We conduct a deeper investigation into its architecture (based on the official implementation in the `transformers` library) and attribute this difference to two key architectural designs:
>
> - 1. **Consistent Positional Encoding:** Qwen2.5-VL employs **Rotary Position Embeddings (RoPE) in both its vision encoder and its language model**. This ensures that the precise spatial information learned by the visual tokens is seamlessly understood and utilized by the LLM. In contrast, LLaVA uses a CLIP vision encoder with a separate positional awareness mechanism, meaning the positional context of its visual tokens must be "re-aligned" within the LLM, making it more sensitive to token pruning that might disrupt this spatial understanding.
> - 2. **Inherent Token Compression:** The Qwen2.5-VL vision encoder contains `PatchMerger` modules, which perform an initial stage of token merging and compression before the features are passed to the LLM. This means that Qwen2.5-VL is **naturally adapted during its training to form representations from a reduced number of tokens**, thus exhibiting greater tolerance to further, explicit pruning.

---

> ### Author Response · Authors · 2025-11-20
> **Response 3**
>
> **Q3: Has any failure analysis been conducted?**
> ##  R3: Case Study
>
> We sincerely thank the reviewer for this insightful suggestion. Since the work on **Vision token pruning is model-agnostic and task-agnostic**, conducting a comprehensive failure case analysis is highly challenging. This difficulty arises because different models possess varying capabilities and different tasks exhibit distinct sensitivities, which is why the vast majority of **previous methods have not conducted such an analysis (e.g. [10]~[14]).**
>
> To address your concern, we have incorporated **a new, dedicated "Case Study" section in Appendix E (Page 28)** of our revised manuscript. We perform targeted case studies based on the premise of our work. Specifically, addressing **the issue of visual grounding performance collapse** caused by token pruning. The analysis is broadly divided into three parts:
>
> - 1. Localization failures resulting from the absence of region filtering.  **(Page 29; Figure 15)**
> - 2. How adopting RPME mitigates the localization collapse problem. **(Page 30; Figure 16)**
> - 3. Localization failures caused by misinterpretations from the VLM model itself. **(Page 30; Figure 17)**
>
> ## Reference
>
> - [1] PyramidDrop: Accelerating Your Large Vision-Language Models via Pyramid Visual Redundancy Reduction. (https://arxiv.org/abs/2410.17247) CVPR'25
> - [2] VScan: Rethinking Visual Token Reduction for Efficient Large Vision-Language Models. (https://arxiv.org/abs/2505.22654)
> - [3] GLDS: Global–Local Diversity Selection for Scalable Token Pruning in Vision–Language Models.
> - [4] Token Pruning in Multimodal Large Language Models: Are We Solving the Right Problem? (https://arxiv.org/abs/2502.11501) ACL'25
> - [5] Feather the Throttle: Revisiting Visual Token Pruning for Vision-Language Model Acceleration. (https://arxiv.org/abs/2412.13180) ICCV'25
> - [6] Boosting Multimodal Large Language Models with Visual Tokens Withdrawal for Rapid Inference. (https://arxiv.org/abs/2405.05803) AAAI'25
> - [7] ST3: Accelerating Multimodal Large Language Model by Spatial-Temporal Visual Token Trimming. (https://arxiv.org/abs/2412.20105) AAAI'25
> - [8] Cross-modal Information Flow in Multimodal Large Language Models. (https://arxiv.org/abs/2411.18620) CVPR'25
> - [9] LOOK-M: Look-Once Optimization in KV Cache for Efficient Multimodal Long-Context Inference. (https://arxiv.org/abs/2406.18139) EMNLP'25
> - [10] VisionZip: Longer is Better but Not Necessary in Vision Language Models (https://arxiv.org/abs/2412.04467) CVPR'25
> - [11] SparseVLM: Visual Token Sparsification for Efficient Vision-Language Model Inference (https://arxiv.org/abs/2410.04417) ICML'25
> - [12] PyramidDrop: Accelerating Your Large Vision-Language Models via Pyramid Visual Redundancy Reduction. (https://arxiv.org/abs/2410.17247) CVPR'25
> - [13] VScan: Rethinking Visual Token Reduction for Efficient Large Vision-Language Models. (https://arxiv.org/abs/2505.22654)
> - [14] Feather the Throttle: Revisiting Visual Token Pruning for Vision-Language Model Acceleration. (https://arxiv.org/abs/2412.13180) ICCV'25

---

> ### Comment · Reviewer_MQfG · 2025-11-25
> **Thanks for the rebuttal**
>
> Thank the authors for the detailed rebuttal and thorough experiments. Many of my concerns, especially about the experiments and failure analysis, are fully addressed. I have updated my scores.

---

> > ### Author Response · Authors · 2025-11-26
> > **Thanks for Your Detailed Review**
> >
> > Thank you for your detailed feedback and for taking the time to review the updates. We’re glad to hear that your concerns have been addressed. Please don’t hesitate to let us know if there’s anything else we can clarify.

---

### Official Review · Reviewer_6Jqk · 2025-11-01

**Soundness:** 3
**Presentation:** 3
**Contribution:** 3
**Rating:** 8
**Confidence:** 4

**Summary:**

This paper introduces a novel method for VLM token pruning. The key insight motivating the approach is the recognition that preserving spatial information remains challenging yet is crucial for maintaining performance on grounding tasks after token reduction. The proposed method successfully leverages this insight to achieve superior performance and efficiency across a series of diverse VLM tasks.

**Strengths:**

* This paper offers a comprehensive examination of VLM token pruning, which yields valuable insights regarding the weaknesses of existing methods.
* A novel and intuitive token pruning method is proposed and shown to be effective across a series of tasks.

**Weaknesses:**

* While the authors provide experimental results across diverse tasks, the selection of VLMs is limited. The generalizability and robustness of the proposed method should be confirmed by replicating the key results on a broader range of VLM architectures currently employed in the field.
* Furthermore, several tables and figures are currently too small and visually dense, making them difficult to read and interpret. The authors are encouraged to enlarge these elements and ensure that all text and data points are clearly legible.

**Questions:**

See above.

---

> ### Author Response · Authors · 2025-11-20
> **Response 1**
>
> **Q1: While the authors provide experimental results across diverse tasks, the selection of VLMs is limited. The generalizability and robustness of the proposed method should be confirmed by replicating the key results on a broader range of VLM architectures currently employed in the field.**
> ## R1: Extension to Advanced Models
>
> To address your concern, we **have already conducted new experiments** by applying our method to the **Qwen2.5-VL model (Page 18; Table 13 and Table 14).**Our results confirm that our approach achieves excellent performance on this advanced model as well, which strongly supports the generalizability and scalability of our proposed framework.
>
> | Retain Token| Method | refcoco-testA | refcoco-testB | refcoco+-testA | refcoco+-testB | refcocog-test | Average |
> | :--- | :--- | :--- | :--- | :--- | :--- | :--- | :--- |
> | | **Vanilla** | 92.56 | 85.16 | 89.02 | 79.15 | 87.24 | 100.00% |
> | **75%** | **Nüwa**  |91.76 | 84.37 | 87.98 | 77.18 | 86.87 | 98.8% |
> | **50%** | **Nüwa** |90.04 | 82.85 | 86.74 | 72.65 | 85.49 | 96.4% |
> | **25%** |**Nüwa** | 80.71 | 72.83 | 73.57 | 62.4 | 73.96 | 83.8% |
>
>
> | Retain Token|  Method | GOA | POPE | SQAimg | MMB-en | MME | VQA_text | Average |
> | :--- | :--- | :--- | :--- | :--- | :--- | :--- | :--- |:--- |
> | | **Vanilla** | 61.9 | 87.9 | 77.8 | 83.5 | 2347 | 82.2 | 100.0% |
> | 75%|  **Nüwa** | 60.41 | 87.52 | 77.98 | 83.13 | 2340 | 77.35 | 98.5% |
> | 50%|  **Nüwa** | 59.93 | 87.46 | 78.82 | 83.02 | 2330 | 76.03 | 98.1% |
> | 25%|  **Nüwa** | 58.4 | 87.06 | 78.58 | 82.47 | 2313 | 73.81 | 96.9% |
>
> ---
>
> **Q2: Furthermore, several tables and figures are currently too small and visually dense, making them difficult to read and interpret. The authors are encouraged to enlarge these elements and ensure that all text and data points are clearly legible.**
> ## R2: Readability Improvement
>
> We sincerely thank the reviewer for this valuable feedback. We will reformat the paper for the camera-ready version (which has a 10-page limit) to enhance its readability.

---

### Official Review · Reviewer_DEDu · 2025-11-01

**Soundness:** 3
**Presentation:** 3
**Contribution:** 3
**Rating:** 6
**Confidence:** 4

**Summary:**

This paper presents a two-stage token pruning framework, Nuwa, where the first stage focus on reducing the number of visual tokens by aggregating them via a proximity matrix, and the second state further prune the tokens with text guidance. The proposed Nuwa method is able to reach SOTA performance on multiple VQA benchmarks as well as visual grounding tasks compared to prior methods.

**Strengths:**

- The paper provides a thorough and insightful analysis of the challenges, pitfalls, and properties of token pruning in VLMs.
- The multiple metrics and visualizations in the analysis provides a solid support for the findings.
- The proposed Nuwa is compared with multiple current arts and is able to provide a higher pruning quality that retain the VQA and grounding accuracy with aggressive pruning.

**Weaknesses:**

- It is not clear how those findings have the contributed to the design of the Nuwa algorithm.
- In table 6, the results of FEATHER is only reported for the case of 192 tokens.
- The visual pruning part remain largely depend on the [CLS] token, so it is hard to apply the proposed method on vision backbones without [CLS].
- The text-guided token pruning is pretty strightforward and is not utilizing the prior insights.
- It is a two stage design that requires explicit text-guidance.
- The comparison against Visionzip is not very fair as Visionzip is a text-agnositc method which does not leverage text-guided pruning. It would be better to include the ablaiton study where the first-stage only Nuwa is compared with the Visionzip on several benchmarks.

**Questions:**

See weaknesses.

---

> ### Author Response · Authors · 2025-11-20
> **Response 1**
>
> **Q1: It is not clear how those findings have the contributed to the design of the Nüwa algorithm.**
> ## R1: The design philosophy of the Nüwa
>
> We sincerely thank the reviewer for this insightful and thought-provoking question. It provides us with an excellent opportunity to elucidate **the design philosophy behind our Nüwa algorithm and demonstrate how our research findings progressively led to its final architecture**.
> Our investigation begins with a critical observation of existing visual token pruning methods: the vast majority validate their effectiveness on VQA tasks, while largely overlooking the more demanding Visual Grounding (VG) tasks that require precise localization. As pointed out by recent works, current methods offer limited gains on VQA and can even degrade performance on VG. Our work aims to answer two fundamental questions: **1) Why does this discrepancy exist? 2) How can we systematically solve it?**
>
> Our methodology follows a logical path of **"Identifying the Problem -> Diagnosing the Cause -> Proposing a Solution -> Iterative Refinement,"** as detailed below:
>
> - 1. **Finding #1 (Problem Validation):** We first confirm through extensive experiments that the phenomenon of limited VQA improvement and VG performance degradation is widespread. This convinces us that **improving performance on VG tasks is the most pressing challenge in the field of token pruning.**
> - 2. **Finding #2 (High-Level Diagnosis):** We then investigate how VLMs process visual information, revealing that VQA relies more on semantic understanding, whereas VG is highly sensitive to the precise spatial location of objects. We conclude that existing pruning methods often **disrupt the global coordinate reference system**, thereby impairing localization capabilities.
> - 3. **Design Decision #1 (Region-Aware Pruning):** Motivated by **Finding #2**, we recognize that preserving global spatial information is paramount. Through finer-grained analyses like position reconstruction experiments, we verified that the loss of global context was a key culprit. This directly leads to our **region-aware pruning strategy**. By first dividing the image into regions and then pruning tokens within each, our design **implicitly preserves a coarse-grained global position reference for each token via its parent region.**
> - 4. **Finding #3 (Emergence of a New Challenge):** However, region-based partitioning inevitably retains some low-information or task-irrelevant tokens within each region, which simple pruning cannot effectively eliminate.
> - 5. **Design Decision #2 (Feature Aggregation with Distance Penalty):** To address **Finding #3**, we introduce similarity-based feature aggregation. This, however, presents a new problem: aggregation based solely on feature similarity could erroneously merge tokens that are spatially distant but semantically similar (e.g., two separate cats), thereby **confounding positional information again**. Consequently, we incorporate a **distance penalty** into the aggregation process, prioritizing the merging of spatially proximate tokens. This preserves local spatial structure while consolidating semantic information.
> - 6. **Finding #4 (Deeper Mechanism Insights):** Despite these improvements, performance is not yet optimal. Inspired by and through an analysis of gradient-weighted attention and token L2 norms, we study "register-like" tokens in VLMs **(Page 26; Figure 10)**. We discover that **a subset of high-norm tokens act as task-agnostic "global information carriers"** whose feature distributions are vital for model stability. Aggregation operations are found to "pollute" the features of these critical tokens.
> - 7. **Finding #5 (Multimodal Features Aliginment):** Inspired by prior LLM pruning methods (e.g., [1], [2]), we investigated the internal mechanism of multimodal feature alignment. 2D visualization indicates that this alignment is **largely complete by the model's intermediate layers**, which leads to our two-stage pruning.
> - 8. **Final Design (Two-Stage Framework: "Pillar" and "Aggregate" Tokens):** Based on **Finding #4**, we establish Nüwa's core two-stage framework. We explicitly partition the preserved tokens into two roles: **"Pillar Tokens"** (the identified high-norm tokens), which are kept intact without aggregation to anchor the model's global understanding, and **"Aggregate Tokens,"** which are responsible for collecting and fusing features from their neighbors
>
> In summary, **Nüwa is not a collection of modules but a principled framework derived from a logical research progression**. Each design choice is a direct response to a previously identified problem: from recognizing the VG challenge, to designing region-aware pruning for global context, to introducing constrained aggregation for information density, and finally, to proposing the "Pillar-Aggregate" mechanism to protect critical structural information.

---

> ### Author Response · Authors · 2025-11-20
> **Response 2**
>
> **Q2: In table 6, the results of FEATHER is only reported for the case of 192 tokens.**
> ## R2: Regrading the result of FEATHER
>
> We sincerely thank the reviewer for the meticulous examination of our experimental details in Table 6.
>
> The pruning mechanism of FEATHER differs fundamentally from ours and most other baselines. Instead of pre-processing visual tokens after the encoder, it performs a **one-time pruning operation at a single, specific layer within the LLM decoder**. To ensure a fair comparison against our universal metric of "Average Kept Tokens," a conversion of their unique setup is necessary.
>
> According to the FEATHER paper, two primary configurations are reported:
> - 1. Pruning 75% of tokens at the 4th layer.
> - 2. Pruning 75% of tokens at the 9th layer.
>
> We calculate the equivalent average number of visual tokens processed across all LLM layers for these setups using the following formula :
> $$
> \bar{N} _ {\text{tokens}} = \frac{1}{L _ {llm}} \left( L _ {\text{prune}} \cdot N _ {\text{full}} + (L _ {llm} - L _ {\text{prune}}) \cdot N _ {\text{pruned}} \right)
> $$
>
> Where $L _ {llm}$ is the total number of LLM layers (32), $L _ {\text{prune}}$ is the index of the pruning layer, $N _ {\text{full}}$ is the initial token count (576), and $N _ {\text{pruned}}$ is the post-pruning count (576 * 0.25 = 144).
>
> - Setting 1 (Pruning at Layer 4): Average Tokens = (4 × 576 + 28 × 144) / 32 = 192.
> - Setting 2 (Pruning at Layer 9): Average Tokens = (9 × 576 + 23 × 144) / 32 = 265.5.
>
> In Table 6, we report the result from Setting 1, as its equivalent token count of 192 aligns more closely with the compression rates of the other methods being compared in that particular group, thus ensuring a fair basis for comparison.
>
> **Setting 1**
>
> | Criteria | FLOPs Red | Localization (Avg) | OCID-Ref | RefCOCOg | RefCOCO+ | RefCOCO | Open-Ended VQA (Avg) | TextVQA | GQA | VQAv2 | VizWiz | Challenge Sets (Avg) | POPE | TallyQA | VSR | AI2D |
> | :--- | :--- | :--- | :--- | :--- | :--- | :--- | :--- | :--- | :--- | :--- | :--- | :--- | :--- | :--- | :--- | :--- |
> | φ-R + φuniform (Ours) | 61% | 27.2 | 29.1 | 27.2 | 24.7 | 27.7 | 61.2 | 46.6 | 62.3 | 77.4 | 58.4 | 65.4 | 86.0 | 58.9 | 62.7 | 54.0 |
>
> **Setting 2**
> | Criteria | FLOPs Red | Localization (Avg) | OCID-Ref | RefCOCOg | RefCOCO+ | RefCOCO | Open-Ended VQA (Avg) | TextVQA | GQA | VQAv2 | VizWiz | Challenge Sets (Avg) | POPE | TallyQA | VSR | AI2D |
> | :--- | :--- | :--- | :--- | :--- | :--- | :--- | :--- | :--- | :--- | :--- | :--- | :--- | :--- | :--- | :--- | :--- |
> | φ-R + φuniform (Ours) | 50% | 35.6 | 32.0 | 35.9 | 35.4 | 38.8 | 62.7 | 51.7 | 62.4 | 78.1 | 58.6 | 66.0 | 87.4 | 59.1 | 63.6 | 54.0 |
>
> ---
> **Q3: The visual pruning part remain largely depend on the [CLS] token, so it is hard to apply the proposed method on vision backbones without [CLS].**
> ## R3: The model without [CLS] Token
>
> Our method is designed with this flexibility in mind and can be readily adapted to such architectures. In fact, our newly added experiments on the **Qwen2.5-VL  (Page 18; Table 13, Table 14)**  model have already demonstrated the effectiveness of this adaptation.
>
> For vision backbones that do not incorporate a `[CLS]` token, we employ a widely adopted and effective alternative strategy to derive token importance scores. The rationale is as follows:
>
> - **Rationale:** In [CLS]-based architectures, the attention weights corresponding to the `[CLS]` token reflect the contribution of each visual token to the global image representation. In its absence, the **mean attention score received by a token from all other tokens** serves as a robust proxy for its overall importance within the global context of the feature map.,
> - **Implementation:** Given an attention map $A \in \mathbb{R}^{H \times N \times N}$ (where $H$ is the number of heads and $N$ is the number of tokens), we compute the importance score $S _ j$ for each token $j$ by averaging the attention it receives from all other tokens:
> $$
> S _ j = \frac{1}{N} \sum _ {i=1}^{N} A _ {i,j}
> $$
>   This score vector $S$ then replaces the [CLS]-based importance metric in our original pipeline, while the subsequent pruning steps remain identical.
>
> The success of this approach is validated by our results on Qwen2.5-VL, confirming that our method's efficacy is **not contingent on the presence of a `[CLS]` token** and thus exhibits strong architectural generalizability.

---

> ### Author Response · Authors · 2025-11-20
> **Response 3**
>
> **Q4: The text-guided token pruning is pretty strightforward and is not utilizing the prior insights.**
> ## R4: Regarding  text-guided token pruning
>
> We sincerely thank the reviewer for this insightful comment on our text-guided token pruning module. We will elaborate on the deep motivation and empirical evidence behind this design.
>
> While the design of our text-guided pruning module may appear straightforward, it is **a principled and targeted intervention** rooted in a deep understanding of the internal workings of VLMs. Our design rationale is as follows:
>
> - 1. **Inspiration from Literature:** We are initially inspired by a series of **pioneering works on VLM interpretability (such as [3], [4], [5]).** These studies collectively reveal a critical insight: multimodal information fusion is not an instantaneous event at the exact layer but a **progressive alignment process** that peaks in the middle layers of the model. Introducing text guidance too early or too late would be suboptimal.
> - 2. **Our Empirical Findings:** To pinpoint the optimal moment for intervention, we conduct our own empirical analysis. Specifically, we **visualize the 2D feature distributions of visual tokens and Text Tokens at various depths** (layers 0, 8, 16, 24, and 32), as detailed in **Appendix D.1, Figure 8 (Page 25)**. Our experiments clearly demonstrated that:
>   -- In shallow layers (e.g., 0-8), visual and textual features remain relatively segregated.
>   -- In the middle layers (around 16-24), we observe the most effective alignment and fusion of multimodal features.
> - 3. **Design Choice:** Based on this finding, we choose to apply text-guided pruning at layer 16. This decision considers two factors: **1. Effectiveness:** this stage, the multimodal information is sufficiently fused for the text to serve as a reliable guide for pruning; **2. Efficiency:** Pruning at this intermediate layer maximizes the computational savings for all subsequent layers.
> - 4. **Validation via Ablation:** To prove that this design is not merely "straightforward" but functionally essential, we conduct **ablation studies (Page 9; Table 8)**. We compare our text-guided approach **against a random pruning baseline** implemented at the same layer. The results show that our method significantly outperforms random pruning, which powerfully validates the necessity and effectiveness of using textual information as a guide.
>
>
> In summary, our text-guided pruning module is a deliberate design grounded in solid theoretical inspiration and data-driven insights. It directly translates our key research finding—that optimal multimodal alignment occurs in the middle layers—into a simple yet highly effective algorithmic component.
>
> ## **Reference**:
> - [1] SparseVLM: Visual Token Sparsification for Efficient Vision-Language Model Inference (https://arxiv.org/abs/2410.04417) ICML'25
> - [2] PyramidDrop: Accelerating Your Large Vision-Language Models via Pyramid Visual Redundancy Reduction. (https://arxiv.org/abs/2410.17247) CVPR'25
> - [3] Cross-modal Information Flow in Multimodal Large Language Models（https://arxiv.org/abs/2411.18620) CVPR25
> - [4] Implicit Multimodal Alignment: On the Generalization of Frozen LLMs to Multimodal Inputs (https://arxiv.org/abs/2405.16700) NIPS24
> - [5] Token Activation Map to Visually Explain Multimodal LLMs （https://arxiv.org/abs/2506.23270) ICCV25

---

> ### Author Response · Authors · 2025-11-20
> **Response 4**
>
> **Q5: It is a two stage design that requires explicit text-guidance.**
> ## R5:  Regarding the two stage design
>
> We sincerely thank the reviewer for this valuable feedback. **In fact, our work stems from a comprehensive and systematic analysis, which subsequently led to the design of our two-stage pruning framework.** Viewing the pruning framework from a higher level of abstraction (detailed discussion on Related Work: Page 14), the pruning mechanism can be applied in three stages: 1. Before LLM, 2. Within LLM, 3. After LLM (KV Cache), and combinations thereof. Therefore, **two-stage pruning is not a weakness**. In addition, it is worth noting that popular works, including [2] and [6]~[8], also adopt a two-stage pruning framework.
>
> Specifically, we conduct **a detailed experimental analysis** regarding the problems of token pruning in the visual grounding task-- covering extensive problem validation, analysis of the model's information flow processing, and finally localization of the problem's root cause-- thereby **leading to the proposal of our improved two-stage pruning method.**
>
> Furthermore, we consider **the two-stage method an optimization of the one-stage method**. The one-stage method primarily performs pruning only in the visual-semantic space, but this often fails to effectively utilize the unique characteristics of VLM models, namely, **the alignment of text and visual semantics**. Although the two-stage method introduces some extra computation in the subsequent stage, this is negligible compared to the overall computational workload (Computational analysis in the Appendix C - Page22  Table 17; Prefill time analysis: Page 8 - Table 4).
>
> If you still have concerns regarding the two-stage framework, for example, why you perceive it as a weakness, we welcome further discussion.
>
> - [2] PyramidDrop: Accelerating Your Large Vision-Language Models via Pyramid Visual Redundancy Reduction. -(https://arxiv.org/abs/2410.17247) CVPR'25
> - [6] Global Compression Commander: Plug-and-Play Inference Acceleration for High-Resolution Large Vision-Language Models (https://arxiv.org/abs/2501.05179) AAAI26
> - [7] Multi-Stage Vision Token Dropping: Towards Efficient Multimodal Large Language Model (https://arxiv.org/pdf/2411.10803)
> - [8] VScan: Rethinking Visual Token Reduction for Efficient Large Vision-Language Models (https://arxiv.org/abs/2505.22654)
>
> ---
>
> **Q6: The comparison against Visionzip is not very fair as Visionzip is a text-agnositc method which does not leverage text-guided pruning. It would be better to include the ablaiton study where the first-stage only Nüwa is compared with the Visionzip on several benchmarks.**
> ## R6: Regarding fair comparison with  Visionzip
>
> We sincerely thank the reviewer for this valuable feedback. We ensure fairness in all experimental settings, as **the overall computational load, the pre-fill time and the average number of tokens are nearly identical** (Page 8; Table 4), even though the underlying frameworks may differ.
>
> - 1. Our method leverages the text to **further reduce** tokens, instead of incorporating text features into the tokens through extra computation. This means we utilize fewer tokens in the computation of subsequent layers, thereby eliminating any fairness concerns. Furthermore, if the second-stage pruning were to be removed, our performance would only improve due to the inclusion of more vision tokens.
> - 2. Additionally, we have already performed an **ablation on the second-stage pruning** (see Page 9; Table 8) in our ablation study section by replacing the text-guided pruning with random pruning.

---

### Official Review · Reviewer_rCym · 2025-11-03

**Soundness:** 3
**Presentation:** 3
**Contribution:** 3
**Rating:** 4
**Confidence:** 3

**Summary:**

This paper tackles spatial degradation in vision token pruning for VLMs and proposes RPME to preserve spatial integrity during token reduction. The idea is clear and grounded in practical bottlenecks for high-resolution inference. Experiments across grounding and VQA benchmarks show meaningful gains, especially on spatial tasks, and the ablations strengthen the causal claims.

**Strengths:**

1. Clear motivation grounded in real multimodal deployment constraints.

2. Strong empirical gains on grounding datasets; convincing ablations.

**Weaknesses:**

1. The paper primarily evaluates on LLaVA-1.5-7B (and NeXT-7B), but does not include stronger or more modern VLM backbones such as Qwen-VL, InternVL. Since the field is rapidly standardizing around Qwen-VL and InternVL as competitive baselines, the absence of these comparisons leaves uncertainty about generalizability across architectures.

2. Lack of comparison with stronger baselines, such as PyramidDrop and Vscan.

[1] PyramidDrop: https://arxiv.org/abs/2410.17247
[2] Vscan: https://arxiv.org/abs/2505.22654

3. Limited discussion of failure modes or robustness to cluttered scenes and occlusions.

**Questions:**

See weakness

---

> ### Author Response · Authors · 2025-11-20
> **Response**
>
> **Q1: The paper primarily evaluates on LLaVA-1.5-7B (and NeXT-7B), but does not include stronger or more modern VLM backbones such as Qwen2.5-VL, InternVL. Since the field is rapidly standardizing around Qwen2.5-VL and InternVL as competitive baselines, the absence of these comparisons leaves uncertainty about generalizability across architectures.**
>
>
> ## R1: Extension to Advanced Models (Qwen2.5-VL)
> We sincerely thank the reviewer for this valuable feedback regarding the generalizability of our method. We agree that demonstrating its applicability to more diverse and advanced VLMs is crucial for validating its effectiveness.
>
> * **1. Extension to Advanced Models (Qwen2.5-VL):**
>
> To address your concern, we have **already conducted new experiments** by applying our method to the  **Qwen2.5-VL (Page 18; Table 13, Table 14)** model. Our results confirm that our approach achieves excellent performance on this advanced model as well, which strongly supports the generalizability and scalability of our proposed framework.
>
>
> | Retain Token| Method | refcoco-testA | refcoco-testB | refcoco+-testA | refcoco+-testB | refcocog-test | Average |
> | :--- | :--- | :--- | :--- | :--- | :--- | :--- | :--- |
> | | **Vanilla** | 92.56 | 85.16 | 89.02 | 79.15 | 87.24 | 100.00% |
> | **75%** | **Nüwa**  |91.76 | 84.37 | 87.98 | 77.18 | 86.87 | 98.8% |
> | **50%** | **Nüwa** |90.04 | 82.85 | 86.74 | 72.65 | 85.49 | 96.4% |
> | **25%** |**Nüwa** | 80.71 | 72.83 | 73.57 | 62.4 | 73.96 | 83.8% |
>
>
> | Retain Token|  Method | GOA | POPE | SQAimg | MMB-en | MME | VQA_text | Average |
> | :--- | :--- | :--- | :--- | :--- | :--- | :--- | :--- |:--- |
> | | **Vanilla** | 61.9 | 87.9 | 77.8 | 83.5 | 2347 | 82.2 | 100.0% |
> | 75%|  **Nüwa** | 60.41 | 87.52 | 77.98 | 83.13 | 2340 | 77.35 | 98.5% |
> | 50%|  **Nüwa** | 59.93 | 87.46 | 78.82 | 83.02 | 2330 | 76.03 | 98.1% |
> | 25%|  **Nüwa** | 58.4 | 87.06 | 78.58 | 82.47 | 2313 | 73.81 | 96.9% |
>
> * **2. Implementation Considerations for InternVL:**
>
> We notice that **none of the previous methods (e.g. [1]~[5] and more) included Internvl** within the scope of their experiments. Furthermore, we also perform a careful technical evaluation of implementing our method on InternVL. We identified some **technical challenges** that would hinder fair and robust integration. The primary issue is that the official InternVL codebase does not fully comply with the standard API of the widely-adopted `Hugging Face Transformers` library (see **issue [6]**), with which our method is deeply integrated.
>
> If the results of the InternVL experiment are critical to you, we will make every effort to complete the experiment in the future.
>
> **Reference:**
>
> - [1] VisionZip: Longer is Better but Not Necessary in Vision Language Models (https://arxiv.org/abs/2412.04467) CVPR'25
> - [2] SparseVLM: Visual Token Sparsification for Efficient Vision-Language Model Inference (https://arxiv.org/abs/2410.04417) ICML'25
> - [3] PyramidDrop: Accelerating Your Large Vision-Language Models via Pyramid Visual Redundancy Reduction. (https://arxiv.org/abs/2410.17247) CVPR'25
> - [4] VScan: Rethinking Visual Token Reduction for Efficient Large Vision-Language Models. (https://arxiv.org/abs/2505.22654)
> - [5] Feather the Throttle: Revisiting Visual Token Pruning for Vision-Language Model Acceleration. (https://arxiv.org/abs/2412.13180) ICCV'25
> - [6] https://github.com/huggingface/transformers/issues/33611
>
> ---
>
> **Q2: Lack of comparison with stronger baselines, such as PyramidDrop and Vscan.**
> ## R2 (1/2): Comparison  with VScan and PyramidDrop
>
> We sincerely thank the reviewer for suggesting the comparison with VScan and PyramidDrop.
>
> ###  1. About PyramidDrop
>
> We would like to clarify that **PyramidDrop is indeed one of our key baselines, which we abbreviated as PDrop in our tables for brevity (page 9; Table 5)**. We apologize if this abbreviation caused any confusion.
>
> In particular, regarding the reproduction of its results on the RefCOCO dataset. We observe that the experimental setup of PyramidDrop, particularly concerning the definition of "compression rate" or "average tokens", exhibits certain **inconsistencies and ambiguities** when compared to other baselines like VisionZip [1] and SparseVLM [2]. This issue has also been noted and discussed in the **official repository's issues [7]**.
>
> To ensure a fair and rigorous comparison across all methods, we primarily adhere to the established evaluation protocols and reported results from the recent work, SparseVLM [2] . We believe this approach provides a more reliable and directly comparable performance assessment for our readers.
>
> **Reference:**
>
> - [1] VisionZip: Longer is Better but Not Necessary in Vision Language Models (https://arxiv.org/abs/2412.04467) CVPR'25
> - [2] SparseVLM: Visual Token Sparsification for Efficient Vision-Language Model Inference (https://arxiv.org/abs/2410.04417) ICML'25
> - [7] https://github.com/Cooperx521/PyramidDrop/issues/10

---

> ### Author Response · Authors · 2025-11-20
> **Response 2**
>
> ## R2 (2/2): Comparison with VScan and PyramidDrop
>
> ###  2. About Vscan
>
> Following this invaluable suggestion, we have conducted additional experiments to provide a direct comparison between our method and VScan across several leading Multimodal Large Models.
>
> For a comprehensive evaluation, we provide a **two-fold comparison**: 1) on the exact datasets validated in the VScan paper, and 2) on the broader set of benchmarks used in our paper. The summarized results are as follows:
>
> #### **1. LLAVA-1.5**
>
> - 1. On the exact datasets validated in the VScan：
> | Average Token | Method | Source | GQA | MMB | MMB-Cn | MME | VQAv2 | VQAtext | POPE | SQA | VizWiz | avg |
> | :--- | :--- | :--- | :--- | :--- | :--- | :--- | :--- | :--- | :--- | :--- | :--- | :--- |
> | | **Vanilla** | CVPR'24 | 61.9 | 64.7 | 58.1 | 1862 | 78.5 | 58.2 | 85.9 | 69.5 | 50 | 100% |
> | 192 | **Nüwa** | - | 60.9 | 64.3 | 54.2 | 1834 | 75.9 | 57.4 | 86.4 | 68.2 | 53.3 | 98.91% |
> | 128 | **Nüwa** | - | 60.2 | 63.4 | 53.4 | 1828 | 75.1 | 57 | 85.5 | 67.8 | 53.8 | 98.18% |
> | 64 | **Nüwa** | - | 58.3 | 62 | 51.7 | 1706 | 72.8 | 54.9 | 83 | 67.5 | 55.6 | 95.85% |
> | | | | | | | | | | | | | |
> | 192 | **vscan** | | 60.6 | 63.9 | 57.4 | 1806 | 77.8 | 57.7 | 86.2 | 68.6 | 50.4 | 98.95% |
> | 128 | **vscan** | | 59.8 | 63 | 58 | 1792 | 77.1 | 57.3 | 86.1 | 68.9 | 51.7 | 98.83% |
> | 64 | **vscan** | | 58.3 | 62.1 | 55.7 | 1698 | 75.4 | 55.6 | 85 | 69.2 | 51.8 | 96.77% |
>
> - 2. On the broader set of benchmarks used in our paper:
>
> | Average Token | Method | Source | GQA | MMB | MMB-Cn | MME | VQAv2 | VQAtext | POPE | SQA | VizWiz | MMMU | SEED | avg |
> | :--- | :--- | :--- | :--- | :--- | :--- | :--- | :--- | :--- | :--- | :--- | :--- | :--- | :--- | :--- |
> | | **Vanilla** | CVPR'24 | 61.9 | 64.7 | 58.1 | 1862 | 78.5 | 58.2 | 85.9 | 69.5 | 50 | 36.3 | 58.6 | 100% |
> | 192 | **Nüwa** | - | 60.9 | 64.3 | 54.2 | 1834 | 75.9 | 57.4 | 86.4 | 68.2 | 53.3 | 35.5 | 59.7 | 99.08% |
> | 128 | **Nüwa** | - | 60.2 | 63.4 | 53.4 | 1828 | 75.1 | 57 | 85.5 | 67.8 | 53.8 | 35.8 | 58.7 | 98.40% |
> | 64 | **Nüwa** | - | 58.3 | 62 | 51.7 | 1706 | 72.8 | 54.9 | 83 | 67.5 | 55.6 | 36.4 | 56.44 | 96.29% |
> | | | | | | | | | | | | | | | |
> | 192 | **vscan** | | 60.6 | 63.9 | 57.4 | 1806 | 77.8 | 57.7 | 86.2 | 68.6 | 50.4 | - | - | 98.95% |
> | 128 | **vscan** | | 59.8 | 63 | 58 | 1792 | 77.1 | 57.3 | 86.1 | 68.9 | 51.7 | - | - | 98.83% |
> | 64 | **vscan** | | 58.3 | 62.1 | 55.7 | 1698 | 75.4 | 55.6 | 85 | 69.2 | 51.8 | - | - | 96.77% |
>
> - **Performance on LLaVA-1.5:**
> Our method achieves competitive performance with VScan. On VQA-related datasets, their performance is largely on par. We only observe a minor gap of approximately 1% in a single, specific setting (at 64 average tokens).
>
> #### **2. LLAVA-Next**
>
> | Average Token | Method | GQA | MMB | MMBCN | MME | POPE | SQAIMG | VQAV2 | VQAText | VizWiz | Average |
> | :--- | :--- | :--- | :--- | :--- | :--- | :--- | :--- | :--- | :--- | :--- | :--- |
> | | **Vanilla** | 64.2 | 67.4 | 60.6 | 1851 | 86.5 | 70.1 | 81.8 | 61.3 | 57.6 | 100.00% |
> | 320 | **Nüwa** | 62.3 | 63.2 | 55.9 | 1813 | 86 | 68.2 | 78.5 | 58.5 | 58.94 | 96.83% |
> | 320 | **vscan** | 60.7 | 65.3 | 57.8 | 1767 | 85.1 | 66.9 | 77.1 | 58 | 53.8 | 95.40% |
>
> #### **3. Qwen2.5-VL**
>
>
> | Condition | refcoco-testA | refcoco-testB | refcoco+-testA | refcoco+-testB | refcocog-test | Average |
> | :--- | :--- | :--- | :--- | :--- | :--- | :--- |
> | | **Vanilla** | 92.56 | 85.16 | 89.02 | 79.15 | 87.24 | 100.00% |
> | **0.75** | **Nüwa**  |91.76 | 84.37 | 87.98 | 77.18 | 86.87 | 98.8% |
> | **0.5** | **Nüwa** |90.04 | 82.85 | 86.74 | 72.65 | 85.49 | 96.4% |
> | **0.25** |**Nüwa** | 80.71 | 72.83 | 73.57 | 62.4 | 73.96 | 83.8% |
> | | | | | | | | |
> | **0.75** | **vscan** |91.94 | 83.96 | 87.9 | 74.15 | 86.55 | 98.3% |
> | **0.5** | **vscan** |90.74 | 82.37 | 86.12 | 71.67 | 84.44 | 96.1% |
> | **0.25** | **vscan** |79.05 | 68.22 | 73.72 | 58.95 | 69.43 | 80.7% |
>
> **Performance on Advanced Models (LLaVA-Next, Qwen2.5-VL):**
> - On these more advanced models, our method **demonstrates clear advantages**.
> - On LLaVA-Next: Our method consistently **outperforms VScan by approximately 1%** across all VQA metrics.
> - On Qwen2.5-VL: Our method **surpasses VScan across all settings**, with a particularly significant margin of **3%** under the 25% average token configuration.
>
>
> ### **Analysis and Insights:**
> We attribute this performance difference to the varying image processing strategies employed by these models. LLaVA-1.5 resizes input images to a fixed, lower resolution (e.g., 336 x 336), which can diminish the fine-grained visual details that our feature aggregation method is designed to preserve. **In contrast, newer models like LLaVA-Next and Qwen2.5-VL tend to process images at higher or native resolutions. This allows our method to fully leverage its strength in retaining and utilizing crucial visual information, leading to substantial performance gains.**

---

> ### Author Response · Authors · 2025-11-20
> **Response 3**
>
> **Q3: Limited discussion of failure modes or robustness to cluttered scenes and occlusions.**
> ## R3: Case Study
>
> We sincerely thank the reviewer for this insightful suggestion. Since the work on **Vision token pruning is model-agnostic and task-agnostic**, conducting a comprehensive failure case analysis is highly challenging. This difficulty arises because different models possess varying capabilities and different tasks exhibit distinct sensitivities, which is why the vast majority of **previous methods have not conducted such an analysis (e.g. [1]~[5])**.
>
> To address your concern, we have **incorporated a new, dedicated "Case Study" section in Appendix E (Page 28)** of our revised manuscript. We perform targeted case studies based on the premise of our work, specifically addressing **the issue of visual grounding performance collapse** caused by token pruning. The analysis is broadly divided into three parts:
> - 1. Localization failures resulting from the absence of region filtering.  **(Page 29; Figure 15)**
> - 2. How adopting RPME mitigates the localization collapse problem. **(Page 30; Figure 16)**
> - 3. Localization failures caused by misinterpretations from the VLM model itself. **(Page 30; Figure 17)**
>
> ## **Reference:**
>
> - [1] VisionZip: Longer is Better but Not Necessary in Vision Language Models (https://arxiv.org/abs/2412.04467) CVPR'25
> - [2] SparseVLM: Visual Token Sparsification for Efficient Vision-Language Model Inference (https://arxiv.org/abs/2410.04417) ICML'25
> - [3] PyramidDrop: Accelerating Your Large Vision-Language Models via Pyramid Visual Redundancy Reduction. (https://arxiv.org/abs/2410.17247) CVPR'25
> - [4] VScan: Rethinking Visual Token Reduction for Efficient Large Vision-Language Models. (https://arxiv.org/abs/2505.22654)
> - [5] Feather the Throttle: Revisiting Visual Token Pruning for Vision-Language Model Acceleration. (https://arxiv.org/abs/2412.13180) ICCV'25

---

> > ### Comment · Reviewer_rCym · 2025-11-20
> > **Thanks for the rebuttal.**
> >
> > Thank you for your response. I improved my rating from 4 -> 6.
> >
> > The added Qwen2.5-VL as backbone model makes the paper more solid.
> >
> > Regarding missing baselines, I suggest authors them into revised version.

---

> > > ### Author Response · Authors · 2025-11-20
> > > **Thank You for Your Review**
> > >
> > > Thank you for your detailed feedback and for taking the time to review the updates. We’re glad to hear that your concerns have been addressed. Please don’t hesitate to let us know if there’s anything else we can clarify.

---

### Author Response · Authors · 2025-11-29
**Summary of rebuttal**

**Dear Program Chairs, Senior Area Chairs, Area Chairs, and Reviewers:**

We understand the significant challenges caused by the recent OpenReview incident and appreciate the extra effort required from you to evaluate our submission.

Given that the review scores and discussions have been reverted to their pre-rebuttal state, we would like to provide a factual summary of the constructive exchanges we had with the reviewers **prior to the reversion**.

Before the system reset, our rebuttal and subsequent revisions had successfully addressed key concerns, resulting in a score **improvement from (8, 6, 4, 4) to (8, 6, 6, 6)**. We outline the details of these exchanges below to assist in your assessment:

# 1. Timeline of Events
To demonstrate that the score improvements were the result of standard scientific discourse and occurred prior to the widespread knowledge of the OpenReview bug, we provide the following timeline:

- Nov 11: Rebuttal period started.
- Nov 20: We submitted detailed responses and the revised paper.
- **Nov 20:** Reviewer rCym acknowledged our improvements and **raised their score (4 -> 6).**
- **Nov 26:** Reviewer MQfG acknowledged our clarifications and **raised their score (4 -> 6).**
- Nov 28: ICLR/OpenReview officially disclosed the data leak and announced the score reversion.

# 2. Summary of Resolved Concerns & Score Updates
## Reviewer rCym (Score raised: 4 -> 6)
- **Concern 1 (Generalization)**: Pointed out the lack of modern VLM backbones
  - Resolution: We extended our evaluation to **Qwen2.5-VL 7B [Page 18; Table 13, Table 14]**, demonstrating that our method generalizes effectively across different architectures (LLAVA, LLAVA-Next, and QwenVL).
- **Concern 2 (Baselines)**: Requested comparisons with stronger baselines like PyramidDrop and Vscan.
  - Resolution: We conducted additional experiments comparing our method against Vscan, showing superior performance. PyramidDrop appeared as PDrop in the original paper (the reviewer may not have noticed).
- **Concern 3 (Robustness)**: Noted limited discussion on failure modes.
  - Resolution: We added a detailed case analysis in **[Appendix E (Page 29~30)].**
- Outcome: The reviewer's concerns were addressed with our clarifications, and their score was raised on Nov 20.
## Reviewer MQfG (Score raised: 4 -> 6)
- **Concern 1 (Novelty)**: Questioned the novelty of the two-stage pipeline compared to prior works (citing 9 specific papers).
  - Resolution: We provided a **comprehensive comparison with 9 papers**, explicitly highlighting our unique contributions versus the cited works. We clarified that our specific mechanism and interpretable experiment are distinct from existing two-stage approaches.
- **Concern 2 (Discrepancy)**: Questioned the RefCOCO baseline results.
  - Resolution: This is due to **a misunderstanding by the reviewer**. We clarified the settings, and cited the experimental results from relevant papers to prove consistency.
- Outcome: The reviewer's concerns were addressed with our clarifications, and their score was raised on Nov 26.
## Reviewer 6Jqk (Rating: 8, retained)
- **Feedback**: Praised the paper but requested testing on a broader range of VLMs and improving the legibility of figures/tables.
- Resolution: As requested, we expanded the VLM backbone to Qwen2.5-VL 7B [Page 18; Table 13, Table 14].
## Reviewer DEDu (Rating: 6, retained)
- **Feedback**: Raised questions regarding the connection between findings and design, the reliance on the [CLS] token, and requested fair ablation studies against Visionzip.
- Resolution:
  - 1. We provided a detailed explanation of the design and experiments.
  - 2. We responded and addressed each of the detailed design issues, including text-guided pruning design and applicability to non-[CLS] backbones.
  - 3. The relevant ablation experiments have already been presented in the main text.

The positive consensus was reached well before the official announcement of the leak. We hope this summary provides helpful context regarding the current state of the manuscript and the reviewers' satisfaction with our revisions.

Thank you for your time and fair consideration.

Sincerely,

Authors

---

### Meta-Review · Area_Chair_rAG8 · 2026-01-05

**Summary:**

This paper proposes Nüwa, a two-stage vision token pruning framework for efficient vision–language models, motivated by an analysis showing that existing pruning methods severely degrade visual grounding performance by disrupting global spatial reference frames. The method combines spatial-aware token aggregation after the vision encoder with text-guided pruning inside the LLM, achieving strong VQA performance retention and substantially improved grounding under aggressive token reduction.

Reviewers raised several decision-relevant concerns. These include `the generalization to more modern VLM backbones`, `the lack of explicit comparisons with recent strong token-pruning baselines`, `questions regarding the novelty of the two-stage design relative to prior work`, and `the limited analysis of robustness and failure cases, particularly for visual grounding`. These issues were viewed as important factors in assessing the paper’s generality, competitiveness, and technical contribution.

**Reviewer Concerns:**

**1: [Addressed] Generalization to modern VLM backbones (by: `rCym`, `6Jqk`)**

Reviewers questioned whether the proposed method generalizes beyond LLaVA-style backbones, noting the absence of experiments on more recent and widely adopted VLMs such as Qwen2.5-VL or LLaVA-Next. This raised initial uncertainty about architectural generality.

The rebuttal added experiments on Qwen2.5-VL and LLaVA-Next, demonstrating consistent performance retention on VQA and strong gains on grounding tasks under token pruning. These results directly address concerns about generalization and were explicitly acknowledged by reviewers during discussion.

**2: [Addressed] Missing or insufficient comparison with strong baselines (by: `rCym`, `MQfG`)**

Reviewers noted missing or unclear comparisons with recent competitive token pruning methods, particularly PyramidDrop and VScan, which made it difficult to assess the relative advantages of the proposed approach.

The authors clarified that PyramidDrop was already included (under an abbreviated name) and conducted new, direct comparisons with VScan across multiple models and benchmark settings. The additional results show competitive or superior performance, especially under aggressive pruning and on grounding benchmarks, resolving the concern.

**3: [Addressed] Novelty and distinction from prior multi-stage pruning pipelines (by: `MQfG`)**

One reviewer questioned whether the proposed two-stage pruning framework is meaningfully different from prior multi-stage or hierarchical pruning methods, and whether the contributions extend beyond incremental design choices.

The rebuttal provided a detailed, paper-by-paper comparison clarifying how the proposed method differs in both mechanism and motivation, particularly emphasizing the explicit preservation of global spatial reference frames and the supporting diagnostic analyses. The reviewer indicated satisfaction after clarification.

**4: [Addressed] Robustness and failure mode analysis (by: `rCym`)**

A reviewer pointed out the lack of explicit analysis of failure cases or robustness under challenging grounding scenarios (e.g., cluttered scenes, occlusions), which could limit interpretability of the results.

The authors added a dedicated case-study section analyzing grounding failures caused by token pruning and demonstrating how the proposed spatial-aware design mitigates these issues. This was acknowledged positively in the discussion.

**5: [Partially addressed] Connection between analysis and design choices (by: `DEDu`)**

One reviewer asked for clearer justification of how the empirical analysis (e.g., spatial integrity, positional effects) concretely motivates specific design components, such as reliance on salience scores and text-guided pruning.

The rebuttal provided additional explanations linking the diagnostic findings to the overall design and clarified applicability beyond specific architectural assumptions (e.g., [CLS]-based attention). However, some aspects remain only partially resolved: in particular, the necessity of specific design choices (e.g., the exact form of the salience score and the two-stage decomposition) is not fully isolated from alternative formulations, and the analysis-to-design mapping remains largely qualitative rather than supported by targeted ablations. While these issues do not undermine the empirical effectiveness of the method, they leave room for clearer justification.

**Reviewer Scores:**

* **Reviewer rCym (4 -> 6):** The reviewer initially raised concerns regarding generalization, baseline coverage, and robustness. These issues were addressed through additional experiments and analyses in the rebuttal, and the reviewer indicated a more positive assessment during discussion.

* **Reviewer DEDu (6 -> 6):** The reviewer raised questions regarding the connection between analysis and design choices. While some aspects remain partially clarified, the overall assessment is expected to remain unchanged.


* **Reviewer MQfG (4 -> 6):** This reviewer questioned the novelty of the approach and the adequacy of baseline comparisons. Following clarification and expanded experimental results, the reviewer’s assessment became more favorable.

* **Reviewer 6Jqk (8 -> 8):** This reviewer was positive throughout. The additional experiments provided in the rebuttal further reinforced the original evaluation.

---

### Decision · Program_Chairs · 2026-01-26

Accept (Poster)